# Public health implications of satellite-detected widespread damage to WASH infrastructure in the Gaza Strip

**Brian Perlman**[1]*, **Shalean M. Collins**[2], **Jamon Van Den Hoek**[3]

**1** Krieger School of Arts and Sciences, Johns Hopkins University, Washington, DC, United States of America, **2** School of Public Health and Tropical Medicine, Tulane University, New Orleans, Louisiana, United States of America, **3** College of Earth, Ocean, and Atmospheric Sciences, Oregon State University, Corvallis, Oregon, United States of America

* brian@brianperlman.com, bperlma3@jhu.edu

## Abstract

The Israeli military offensive in the Gaza Strip since October 7, 2023, has resulted in widespread attacks across the territory, damaging water, sanitation, and hygiene (WASH) infrastructure. Recent public health assessments show an increased prevalence of water-borne diseases — including polio, Hepatitis A, and gastrointestinal conditions — linked to curtailed access to safe water, exposure to contaminated water, and non-functional WASH infrastructure. However, there is a persistent lack of information on the locations and details of damaged WASH infrastructure across the Gaza Strip that can guide short-term water interventions and inform long-term recovery efforts. This study provides an assessment of the status of Gaza Strip WASH infrastructure through analysis of damage using open-source earth observation and geospatial data. Drawing from six sources, we identified 239 WASH sites spanning 11 types of infrastructure across the Gaza Strip and analyzed very-high resolution satellite imagery at each site to assess indicators of damage incurred through late February 2024. We found that 49.8% (n = 119) of sites had been damaged, including at least half of the desalination plants, water pumping stations, and water towers that formed the backbone of Gaza's WASH infrastructure prior to the escalation of conflict. We observed WASH infrastructure damage in all five governorates, though damage was most pronounced in North Gaza, Gaza, and Khan Yunis Governorates. Due to limited access to multiple sources of satellite imagery, the practical impossibility of creating a comprehensive pre-conflict WASH infrastructure dataset, and the limitation of our scope amid ongoing hostilities, these findings likely represent a conservative under-estimate of total WASH infrastructure damage. While this research does not attribute any individual attack to a specific belligerent, the breadth of WASH infrastructure damage as a result of Israel's invasion in the Gaza Strip points to grave public health consequences, which will have long-lasting repercussions for morbidity and mortality.

**Data availability statement:** The authors are not making the dataset presented in this manuscript publicly available because of the sensitive nature of the data. There are ethical requirements that stipulate that data pertaining to WASH infrastructure locations in an active conflict area cannot be shared; additionally, specific partners have shared vector data with the authors of this study with the stipulation that it not be publicly shared. However, portions of the study dataset are openly available via online open sources and may be made available upon reasonable request on a case-by-case basis. Please contact Cory Langhoff via email at cory.langhoff@oregonstate.edu with detailed information regarding the data requested, such as how it will be stored, used, and distributed. The SkySat satellite imagery utilized in this study was provided by Planet Labs PBC and published with written consent. The authors are not authorized to share this raster data externally per the company's policy; however, data access requests may be sent to Planet Labs PBC via email at images@planet.com and will be handled by the company on an individual basis. The authors of this paper did not have any special access privileges that other researchers would not have.

**Funding:** The authors received no specific funding for this work.

**Competing interests:** The authors have declared that no competing interests exist.

## Introduction

Water insecurity poses serious threats to nutrition, economic productivity, and physical and mental health. Suboptimal water, sanitation, and hygiene (WASH) conditions contribute to 60% of all diarrheal deaths and more than 5% of all deaths in children under five years of age globally [1]. Water insecurity contributes to dehydration, undernutrition (e.g., stunting, wasting), adverse changes to the gut microbiome, stress, and adverse birth outcomes, among other consequences [2]. It can also exacerbate anxiety and depression [3–5], potentially compounding mental health issues that are already heightened in contexts with ongoing conflict and insecurity [6]. Water collection is disproportionately carried out by women [7,8], which can lead to injury [9,10] and increased risk of gender-based violence [11]. Damage to infrastructure necessary for storage, transmission, and provision of WASH services not only limits access to safe water, creating elevated risks of preventable diseases [12], but can also result in environmental degradation [13], including from wastewater leaching into soil, affecting ground and surface water [14,15] or pollution of marine ecosystems from direct discharges into water bodies [15–17].

In armed conflicts, WASH facilities and infrastructure are often targeted by conflict actors or armed groups [18], and the degradation of WASH services through conflict-induced damage or disrepair are a major cause of indirect civilian mortality [19,20]. Conflict effects on water infrastructure can be grouped into four categories [21]: 1) damage or destruction of existing water infrastructure; 2) obstacles to damaged infrastructure repair; 3) limitations on new infrastructure construction; and 4) development of informal infrastructure. Of these, infrastructure damage is perhaps the most well documented because of the prevalence of attacks on WASH infrastructure across so many armed conflicts [22]. For example, in Syria, political unrest gave way to targeted attacks on water systems because of their strategic value [23]. In Ethiopia, violence in the Tigray region "led to considerable damage" to the region's water infrastructure, exposing millions of people to water-related health risks [24]. The effects of armed conflict on WASH infrastructure have also been highlighted in Russia's offensive on Ukraine, where dams have been routinely targeted and damaged, surface water has been polluted, and wastewater treatment facility operations have been disrupted [25]. In the Gaza Strip, damage to WASH infrastructure has been a recurrent strategic approach by the Israeli military since the first Palestinian uprising in 1987 [26,27].

Prior to the escalation of hostilities in the Gaza Strip on October 7, 2023, safe water access was already limited, driven by a lack of surface water sources and asymmetric control of water resources by Israel [28,29]. Only 6% of the population had access to "safely managed, piped drinking water" [15], and over one third (35.2%) of individuals reported experiencing moderate-to-high levels of water insecurity in 2021 [30]. Post October 7, water accessibility has deteriorated considerably [15,27,31–35], and civilians have struggled to meet minimum water needs (i.e., 15 liters per person per day based on the Sphere Humanitarian Minimum Charter [36] — less than one standard-sized jerrycan). By late May 2024, inadequate access to clean water and sanitation resulted in roughly a quarter of the population in the Gaza Strip contracting easily preventable diseases [37].

Previously undocumented levels of water-related health concerns such as dehydration, intestinal disorders, skin rashes, and influenza have been identified since October 2023 [38], with 136,400 cases of diarrhea in children under-five reported in the first three months following conflict escalation [15]; this is in stark contrast to the approximately 918 cases reported across both the Gaza Strip and West Bank in the two weeks prior to the 2019 UNICEF Multiple Indicator Cluster Survey (MICS) (i.e., a baseline prevalence) [39]. Hepatitis A, typically acquired by consuming contaminated food or water or being in contact with an infected person, began spreading in the Gaza Strip in January 2024 according to the World Health

Organization (WHO), and several thousand people have presented with jaundice, a symptom of infection [40]. On August 16, 2024, the Palestinian Ministry of Health confirmed the first case of vaccine-derived polio, a disease primarily transmitted person-to-person via the fecal-oral route (exposure to sewage and contaminated water increases risk of acquisition) [41], ending a 25-year period in which the Gaza Strip was polio-free [42]. At the time of writing, a mass vaccination campaign had administered both doses of the oral polio vaccine (nOPV2) to 94% of the target population of children under the age of 10 in the Gaza Strip [43,44]. Although at least two vaccine doses and a minimum of 90% coverage are needed to stop circulation of polio, coverage in northern Gaza was only approximately 88% due to lack of access [44].

Given the mounting concerns of water-related health risks in the Gaza Strip from global public health officials [45], it is important to understand the location and timing of damage to WASH infrastructure to both improve the chances of preventing the most dangerous health effects of contaminated water exposure and consumption and assess repair and reconstruction needs. Remote, satellite image-based assessments of WASH damage in the Gaza Strip from intergovernmental organizations (e.g., United Nations Satellite Centre (UNOSAT) [46,47] and the European Union [48]), international financial institutions (e.g., World Bank [47,48]), and journalism organizations (e.g., BBC [49]) point to stark rates of infrastructure damage but lack methodological transparency, making validation of the results, evaluation of methods, or review of the underlying data impossible.

Here, we present an accounting of the extent and location of WASH infrastructure damage in the Gaza Strip in the first five months after the escalation of hostilities on October 7, 2023. We use open-source geospatial data on 239 locations of WASH infrastructure, including water treatment plants, storage infrastructure, and wells, across the Gaza Strip and produce site- and type-specific damage assessments through visual examination of very-high resolution (VHR) optical satellite imagery collected from October 8, 2023, to February 22, 2024. We also gauge agreement between our findings and other publicly available assessments of WASH damage and discuss the implications of the results in light of field-based reporting of public health outcomes. This assessment of WASH damage across the Gaza Strip will inform decisions and targeted interventions not only in the WASH sector, but also among public health professionals, academics, and governments/intergovernmental organizations providing life-saving humanitarian assistance and initiating post-conflict reconstruction.

## Methodology

### Collating WASH infrastructure site datasets

We created a database of WASH infrastructure locations in the Gaza Strip by drawing from six sources (Table 1 and Fig 1). Data sources were chosen based on their proximity to the primary (originating) data source, as well as recency, comprehensiveness, data availability, and sharing protocols. All of the data were open-source (available either via observation or request [54]), however some of the data obtained are proprietary and were provided by organizations with the condition that they not be shared externally (i.e., data from the WASH Cluster and the Coastal Municipalities Water Utility).

We identified an initial set of 269 WASH locations across all data sources; 203 WASH infrastructure locations were sourced from a single data source, while 66 locations were sourced from two or more data sources. In the case of duplicates, we selected a single representative location and noted the source datasets. The names of each WASH infrastructure location and the type of WASH infrastructure were recorded from the source datasets in English or translated from Arabic. In the case of disagreement in site name across two or more datasets, the respective names from each dataset were included in our study dataset with proper citation. If two types of

**Table 1. WASH infrastructure data sources.**

| Dataset | Description | Attribute data | Date(s) | Publicly accessible | WASH infrastructure location or damage data | Number of WASH locations |
|---|---|---|---|---|---|---|
| CMWU | Locations of water wells and water tanks. | Names and locations only | Accessed June 2024 | No | Location | 385 |
| OSM [50–52] | OSM data were sourced from three sources, which contained location data on the following infrastructure types:<br>1) Water tanks, water pumps, wastewater treatment plants, water towers, water wells, toilets, and fountains [50].<br>2) Desalination plants, water wells, car washes, fountains, CMWU office building, water pumps, water tanks, and water towers [51].<br>3) Water wells [52]. | 1) Names (for some sites) and locations only<br>2) Names, amenity types, locations, and OSM_IDs<br>3) Names, infrastructure types, locations, and OSM_IDs | 1) Accessed June 2024<br>2) Accessed April 2024 (modified March 2024)<br>3) Accessed February 2024 | Yes | Location | 347, comprised of:<br>1) 252 points<br>2) 91 points<br>3) 4 points |
| UNO-SAT [46] | Locations of damaged or destroyed buildings, including WASH infrastructure. | Names (for some sites), OBJECTIDs, locations, and damage statuses | Published March 2024 | Yes | Location and damage | 59 |
| PAX | Locations of water treatment plants, water pumps, water towers, desalination plants, and reservoirs/lagoons. | Names (for some sites), notes, infrastructure types, and locations | Accessed February 2024 | No | Location | 21 |
| WASH Cluster | Locations of desalination plants. | Names of organizations, infrastructure types, and locations | Accessed July 2024 | No | Location | 2 |
| France24 [53] | Locations of a water tower and a desalination plant. | Locations | Published December 2023 | Yes | Location | 2 |

CMWU, Coastal Municipalities Water Utility; OSM, OpenStreetMap; HOTOSM, Humanitarian OpenStreetMap; HDX, Humanitarian Data Exchange; UNOSAT, United Nations Satellite Centre.

water infrastructure were noted at one location, a combined type was recorded (e.g., water well/ water tank or desalination plant/water well). If more than two types of WASH infrastructure were noted at a location, it was categorized as "multiple." If the infrastructure type was ambiguous in the source data (e.g., only the name of a water company was provided), it was labeled "unknown type." We did not independently verify the names or facility types in the source datasets, although we did inspect pre-October 2023 satellite imagery and online open-source information (e.g., OpenStreetMap (OSM), Wikimapia, Google Maps) to evaluate the source dataset's accuracy, noting any discrepancies. In the case of WASH infrastructure sites that were present in multiple data sources, we also compared source datasets against one another, noting where they agreed and disagreed. Finally, WASH locations were either represented as a point (latitude-longitude) or a polygon, in which case the coordinates of the polygon's centroid location were recorded.

The most common WASH infrastructure types in our study dataset (n = 269) were water wells (29.4%, n = 79), desalination plants (21.2%, n = 57), and water tanks (11.5%, n = 31) (Fig 2). Desalination plants are ubiquitous and critical in the Gaza Strip, due to a lack of surface water which necessitates heavy reliance on salinated water from the Coastal Aquifer and seawater; desalination plants produce clean water from brackish groundwater or seawater for drinking, personal hygiene, and domestic use. Often, desalination plants were small-scale in terms of size and water volume output, and many were reportedly located inside civilian infrastructure, such as schools and mosques. Prior to the escalation of hostilities, groundwater extraction accounted for 81% of the total water available to Palestinians living in Gaza [27]. Groundwater over-extraction contributed to water pollution and salinization of the Coastal Aquifer [31], and approximately 96–97% of the water from this source did not meet WHO standards prior to the conflict [55–57]. Water treatment plants and pumping stations play a crucial role in disinfecting

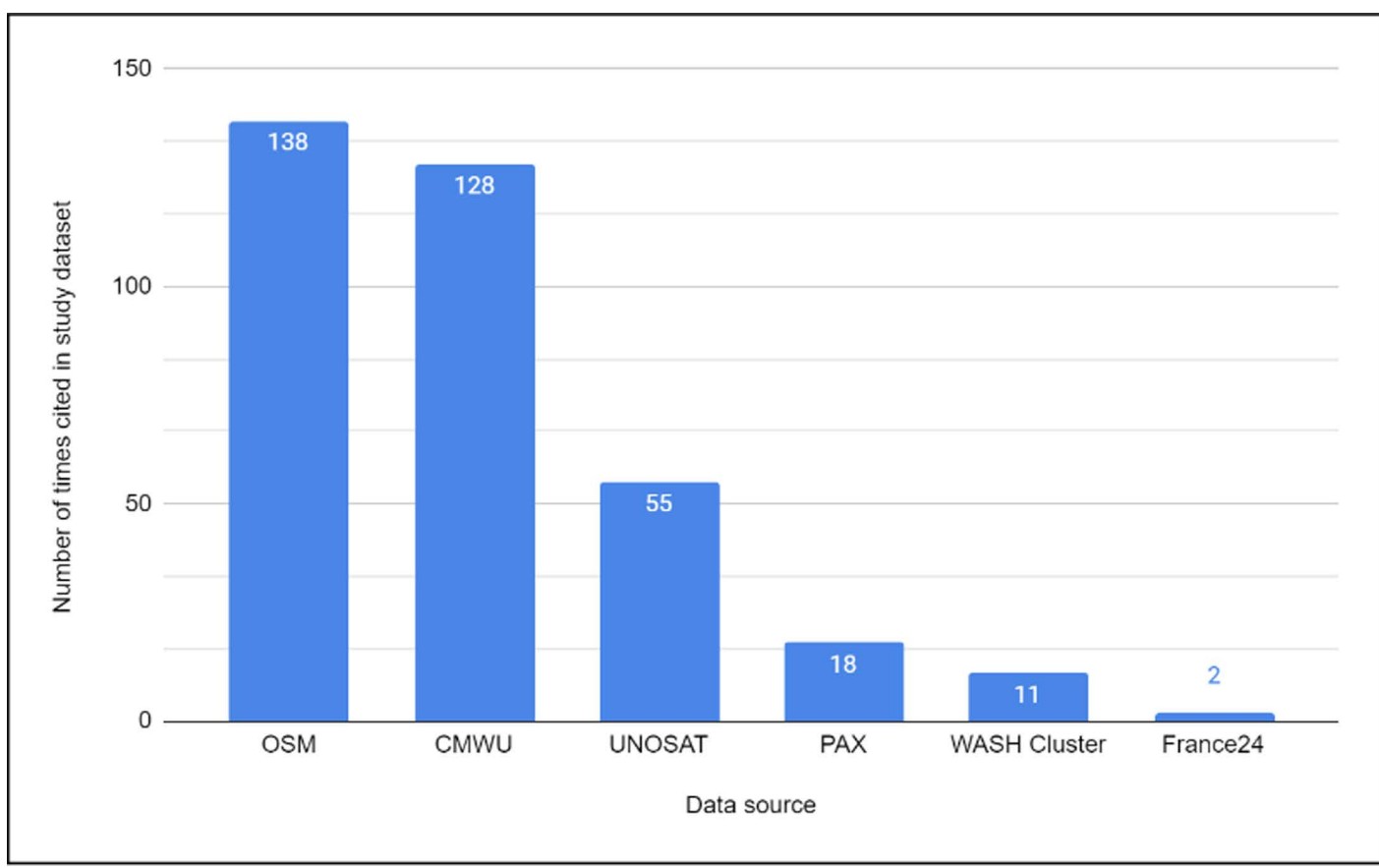

**Fig 1. Count of the number of times a data source was cited in the study dataset.** OSM, OpenStreetMap; CMWU, Coastal Municipalities Water Utility; UNO-SAT, United Nations Satellite Centre.

wastewater and preventing it from overflowing into the urban or natural environment. They facilitate the transfer and safe processing of sewage, which, if left untreated or stagnant in communities, can become a biohazard leading to harmful — and in some cases, fatal — public health outcomes [14,15,58,59]. Although often overlooked, warehouses that store repair equipment and spare parts are valuable assets that if damaged (along with their contents) could prevent authorities from making vital repairs necessary to keep WASH facilities operational. Water storage facilities, such as tanks and towers, were spread throughout communities in the Gaza Strip, providing additional storage capacity for clean water. In our dataset, water wells were sometimes co-located with water storage facilities, offering local safe water access. Our data also included the Coastal Municipalities Water Utility (CMWU) headquarters building, which would have been important for the administration of operating and maintaining WASH facilities in the Gaza Strip, including personnel, funding, and administrative tasks. Additionally, a small number of car washes were included since they could serve as emergency water sources.

## Damage assessment of WASH infrastructure using very-high resolution satellite imagery

We examined Planet Labs' SkySat very-high resolution (VHR) optical satellite imagery captured between May 24, 2021, and June 5, 2023, before the escalation of hostilities, as well as imagery captured during the conflict from October 8, 2023, to February 22, 2024 (i.e., the

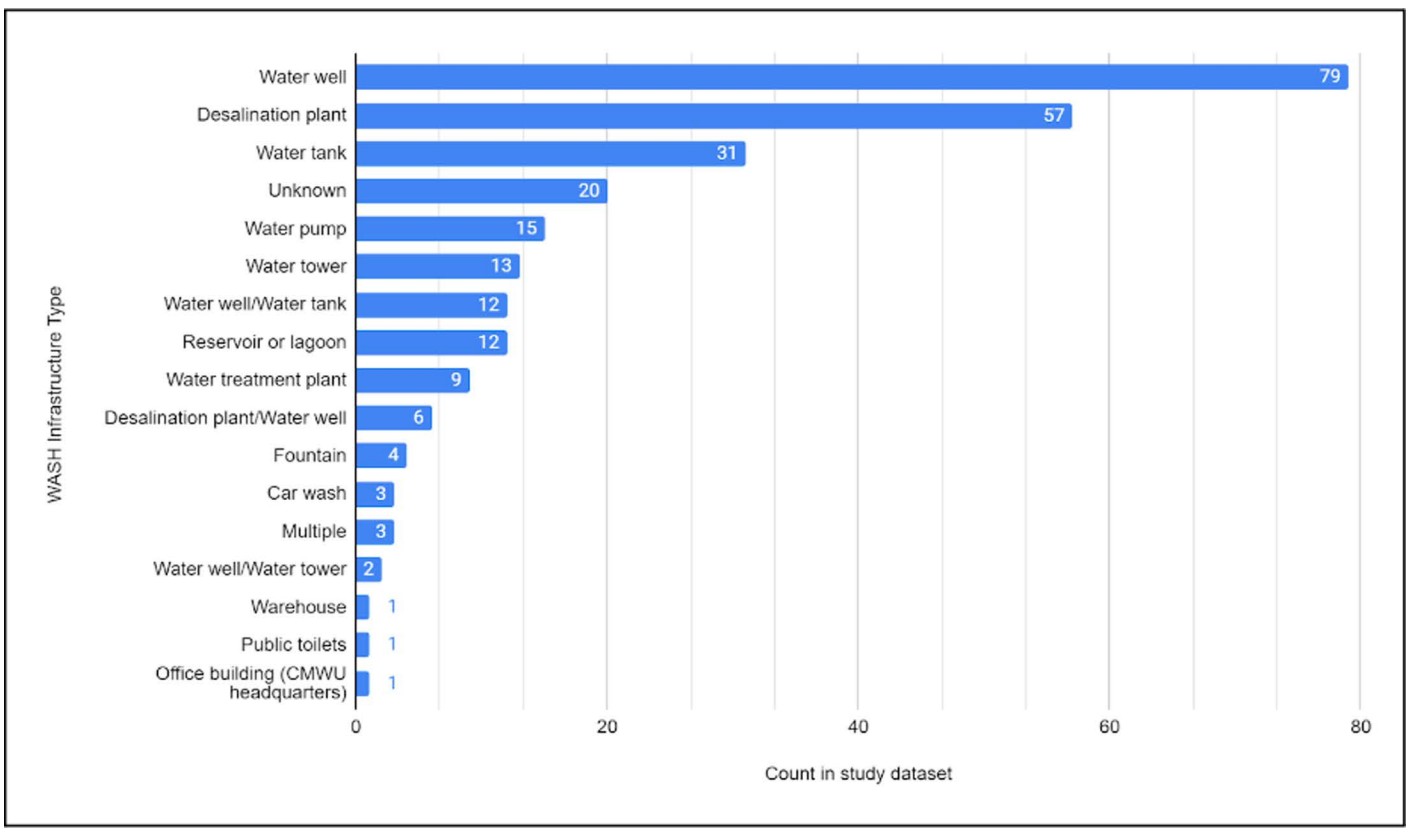

**Fig 2. Count of WASH infrastructure type in the study dataset (n = 269).** CMWU, Coastal Municipalities Water Utility.

study period). Satellite images with a spatial resolution of better than 1 meter ground sample distance (GSD) are often referred to as VHR and typically provide sufficient visual detail for assessing the presence and/or condition of individual buildings and infrastructure. We downloaded Planet Labs' satellite imagery via a Google Drive repository made accessible to humanitarian researchers and journalists and analyzed the imagery in ArcGIS Pro (Version 3.3 series). For pre-conflict imagery, we visually examined each site in the SkySat imagery to ensure that the WASH infrastructure was present before the escalation of conflict; we also used Airbus Pléiades imagery accessible via Google Earth Pro in this pre-conflict assessment, as necessary. To assess damage during the conflict, we visually compared the condition of the site in imagery collected before and during the conflict [60]. We typically chose to analyze several of the most recent images available for a given location to evaluate the presence of damage but also considered imagery collected at different points in the study period to better understand the WASH infrastructure location and its context. A WASH site was classified as *damage visible* if there was any visual indication of damage up to and including apparent destruction. We did not distinguish between different levels of damage severity, e.g., from moderate damage to totally destroyed, to simplify the damage labeling process.

We followed photo-interpretation best practices by orienting images according to the off-nadir angle by referencing buildings with clear sides visible, rotating the image such that "up is up" — orienting the image in the way that the sensor acquired the image — as opposed to "north up" (i.e., rotating the image such that north is at the top of the view window), and observing the position, direction, and length of shadows. Examining multiple images with varying sensor azimuth and off-nadir angles allowed us to see different sides of WASH sites,

ultimately providing a more thorough damage assessment site by site. Examining multiple images of the same location over time also provided a range of illumination conditions that provided more opportunities to assess a site without obstruction due to shadowing.

Damage assessments were assigned a Confidence Interval according to the approximate probability that the site was damaged based on visual interpretation of the satellite imagery. This determination was based on building damage assessment guidelines from the International Working Group on Satellite-based Emergency Mapping [61] and Confidence Interval terminology adapted with permission from Armament Research Services Pty. Ltd. (Table 2). This terminology was used both in determining the validity of a claim that a WASH site existed at a purported location and in our later visual analysis of satellite imagery of vetted WASH locations.

If image quality was compromised by natural phenomena (e.g., clouds, shadowing, haze), errors in image production, or technical inadequacies that precluded interpretation, it was not used for damage assessment. Similarly, if a WASH infrastructure site (or a potentially damaged area of the site) was too small to assess accurately due to the spatial resolution of the available satellite imagery, we did not assign a damage status to that site. In our initial dataset of 269 WASH infrastructure sites, 11.2% (n = 30) were excluded from analysis due to insufficient image quality/interpretability or insufficient confidence in the assessment. Water wells comprised nearly half (n = 14) of the excluded sites due to inconsistent physical structure, small footprint, and often nondescript visual appearance which made it difficult to distinguish nearby damaged areas from potentially damaged well structures.

After we assessed the damage status at the remaining 239 WASH infrastructure sites, we grouped the sites into two categories: 1) *Damage visible* or 2) *No damage visible*. We then summarized the distribution of damaged WASH infrastructure sites by administrative region (i.e., governorate), infrastructure type, and date of first damage detection.

## Results

### Geography and timing of WASH infrastructure damage

Nearly half (49.8%, n = 119 of 239) of WASH infrastructure locations was damaged during the study period (Fig 3). Sixty-three percent of the damaged WASH sites were located in the two northernmost governorates. In North Gaza Governorate, of the 51 sites in our dataset for which we had interpretable imagery, 78.4% (n = 40) were damaged, which comprised 33.6% of all damaged WASH locations across the Gaza Strip. In Gaza Governorate, of the 57 sites in our dataset for which we had interpretable imagery, 61.4% of WASH sites (n = 35) were damaged, which comprised 29.4% of all damaged WASH locations. Damage was more evident in the north, but we detected damaged WASH infrastructure in each governorate, including 52.8% of

**Table 2. Confidence interval terminology.**

| Confidence interval | Appropriate terms and descriptions | Approximate probability range |
|---|---|---|
| CI-0 | Insufficient information; unknown; unable to assess; possible. | Unknown or <10% |
| CI-1 | Low confidence; unlikely. | 10–25% |
| CI-2 | Plausible; feasible. | 26–50% |
| CI-3 | Moderate confidence; likely; probable. | 51–75% |
| CI-4 | High confidence; highly likely. | 76–95% |
| CI-5 | Almost certainly; near-certainty; statement of fact. | ≥96% |

Confidence interval terminology adapted from Armament Research Services Pty. Ltd. (With permission).

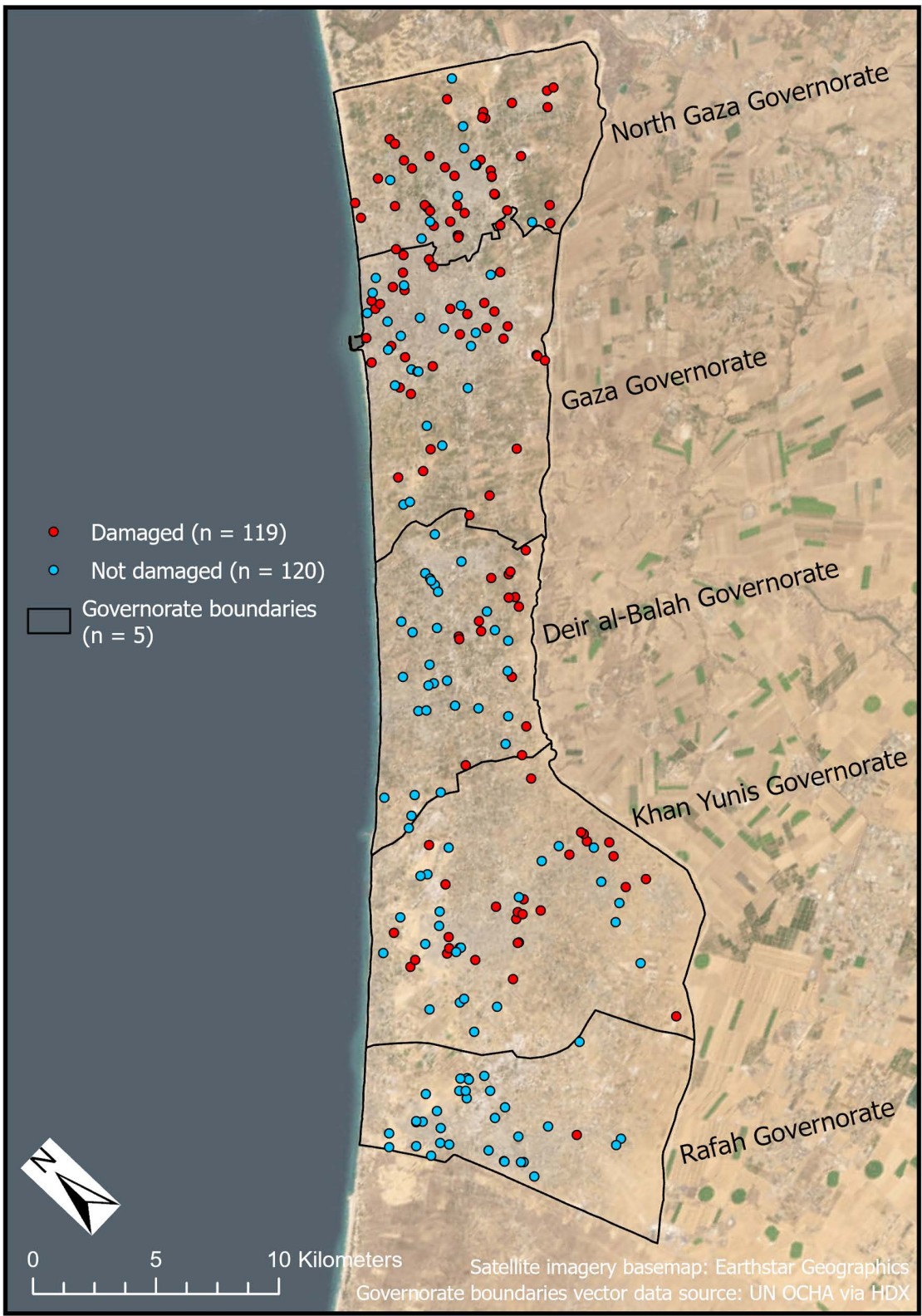

**Fig 3. Map of WASH infrastructure sites assessed for damage (n = 239).** Red points indicate damaged infrastructure (n = 119) and blue points indicate non-damaged infrastructure (n = 120). WASH infrastructure locations that were not assessed for damage due to image interpretability issues (n = 30) are not shown. Governorate boundaries vector data source: UN OCHA via Humanitarian Data Exchange. Satellite imagery basemap: Earthstar Geographics via ESRI.

locations in Khan Yunis Governorate (n = 28 of 53), 34.1% of locations in Deir al-Balah Governorate (n = 15 of 44), and a single site in Rafah Governorate (2.9%, n = 1 of 34) (Table 3). Damage largely occurred earlier in the northern parts of the Gaza Strip, while later damage tended to be closer to the southern part of the Gaza Strip, though not exclusively (Fig 4). These results align with broadly held understandings of the geography and timing of damage since October 2023. Concentrations of urban damage had been reported in the northern areas of the Gaza Strip (i.e., North Gaza and Gaza Governorates) in the first months of the war [46,62,63]. We also expect to see a limited number of damaged WASH infrastructure sites in Rafah Governorate since our study period ended in late February 2024, before much of the reported damage in the governorate [64].

## WASH infrastructure damage by type

All *desalination plant/water well* sites were found to be damaged, as were all *multiple* sites, where three or more infrastructure types were present at a WASH site (Table 4 and Fig 5). Two-thirds of *water pump* sites, *water well/water tank* sites, and WASH sites whose precise type could not be ascertained with sufficient levels of confidence (i.e., *unknown* type) were damaged, while the majority of *desalination plant* sites and *water tower* sites were also damaged. Below, we discuss in detail damage to water desalination plants, water pumping facilities, and water storage infrastructure (i.e., water tanks and water towers).

All of the desalination plants in North Gaza Governorate, half of the desalination plants in Gaza Governorate, roughly a third (31.3%) of the desalination plants in Deir al-Balah Governorate, and over three-quarters (78.6%) of desalination plants in Khan Yunis Governorate for which we had interpretable imagery were assessed as damaged. In Rafah Governorate, there were only two desalination plants in our dataset (both located on the grounds of schools), and both were assessed to be undamaged. Fig 6 shows the Ayyah Well Desalination Plant at various sensor azimuth and off-nadir angles to visualize the different parts of the facility that were damaged. This WASH site, located in a dense urban area of Khan Yunis Governorate, was represented in several source datasets, so it can be assumed that it was a well-known WASH location, both among Palestinians living in Gaza and the conflict belligerents. In addition to the desalination plant, there were also reportedly two water wells located here, according to the Coastal Municipalities Water Utility. We assessed with high confidence that this site sustained conflict damage.

Damage to water pumping facilities was observed primarily in Gaza Governorate, comprising 62.5% of damaged pumping facilities, but damage to water pumps was also observed in North Gaza and Khan Yunis Governorates. Damage was observed at both the Zahra sewage pumping station (Fig 7) and the Bani Suhaila sewage pumping station (Fig 8), located in Gaza Governorate and Khan Yunis Governorate, respectively.

Table 3. WASH infrastructure damage by Gaza Strip governorate.

| Governorate | Number of damaged WASH sites (total number of sites) | Percent damaged WASH sites in governorate | Percent of total damaged WASH sites in Gaza Strip |
|---|---|---|---|
| North Gaza | 40 (51) | 78.4% | 33.6% |
| Gaza | 35 (57) | 61.4% | 29.4% |
| Deir al-Balah | 15 (44) | 34.1% | 12.6% |
| Khan Yunis | 28 (53) | 52.8% | 23.5% |
| Rafah | 1 (34) | 2.9% | 0.8% |
| **Total** | **119 (239)** | **49.8%** | |

Distribution of the number and percentage of damaged and undamaged WASH infrastructure sites by Gaza Strip governorate.

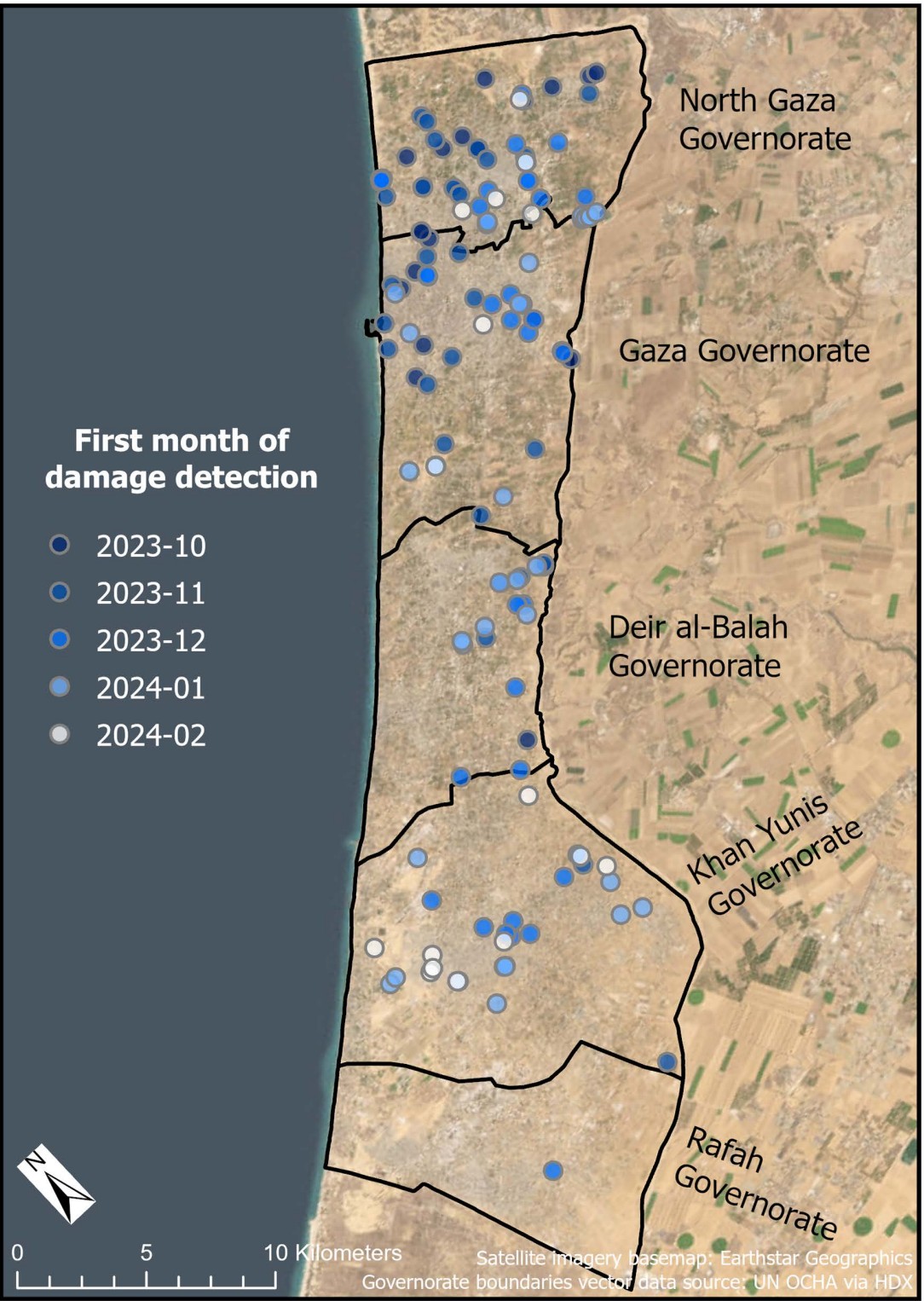

**Fig 4. Map of the first month of damage detected at WASH infrastructure sites.** Earlier damage is symbolized in darker tones, while later damage is symbolized in lighter tones. Governorate boundaries vector data source: UN OCHA via Humanitarian Data Exchange. Satellite imagery basemap: Earthstar Geographics via ESRI.

**Table 4. WASH infrastructure damage by type.**

| WASH infrastructure type | Count of damaged sites | Percent damaged | Percent of total damaged infrastructure |
|---|---|---|---|
| Desalination plant/Water well (n = 6) | 6 | 100% | 5.0% |
| Multiple (n = 3) | 3 | 100% | 2.5% |
| Warehouse (n = 1) | 1 | 100% | 0.8% |
| Unknown (n = 18) | 12 | 66.7% | 10.1% |
| Water well/Water tank (n = 12) | 8 | 66.7% | 6.7% |
| Water pump (n = 12) | 8 | 66.7% | 6.7% |
| Water tower (n = 13) | 7 | 53.8% | 5.9% |
| Desalination plant (n = 49) | 26 | 53.1% | 21.8% |
| Water well (n = 65) | 31 | 47.7% | 26.1% |
| Water treatment plant (n = 9) | 4 | 44.4% | 3.4% |
| Water tank (n = 29) | 9 | 31.0% | 7.6% |
| Reservoir or lagoon (n = 12) | 3 | 25.0% | 2.5% |
| Fountain (n = 4) | 1 | 25.0% | 0.8% |
| **TOTAL** | **119** | | **100%** |

Percent of damage to each type of WASH infrastructure across the Gaza Strip.

Water storage infrastructure, such as tanks and towers, were also damaged, primarily in North Gaza and Khan Yunis Governorates. A collapsed water tower, located adjacent to the Umar bin Abd Al-Aziz Mosque, was observed in Rafah Governorate (Fig 9). We also noted damage to solar panels, which serve as power sources, associated with various water tanks; Maan and Maan New water tanks and water wells (Fig 10) and a BWSS (bulk water supply scheme) water tank (Fig 11) are examples depicted here. Our damage assessment methodology includes damage to features (such as solar panels) on top of a WASH facility, as is the case with the Maan New water tank (north tank) (Fig 10). The Maan water tank (south tank) also shows visible damage to its structure on the northeast corner of its roof; the damaged solar panel infrastructure directly to the southwest of the Maan water tank can only be assumed to be associated with the tank, although it likely is.

Wide-area destruction results in the leveling of a landscape in a specified area and was observed in our analysis. Case examples of wide-area destruction can be seen in the Al Qarara BWSS water tank (Fig 12) and two BWSS locations (Fig 11), in which the immediate vicinities of the WASH infrastructure locations were severely damaged. Damage patterns such as those observed here imply that damage to WASH infrastructure may have resulted from attacks dispersed over a wide-area, as opposed to direct targeting of specific infrastructure locations, although targeting of an area could be influenced by the presence of WASH infrastructure.

## Confidence of damage assessment

Almost all damaged sites (98.3%) were assigned a damage Confidence Interval of *probable* (51–75%) based on interpretation of VHR optical satellite imagery. However, there were a few exceptions (1.7% of damaged sites) in which it was *highly likely* (76–95%) or a *near-certainty* (≥96%) that a WASH infrastructure site was damaged, such as when fire at the WASH site was evident in the imagery (e.g., Fig 13). No sites that were assigned a damage status (either *damage visible* or *no damage visible*) received a Confidence Interval of *plausible* (26–50%), *low confidence* (10–25%), or *possible* (<10% or insufficient information to assign a probability range).

The Bureij Central Wastewater Treatment Plant (a.k.a. Gaza Central Treatment Plant) [65] was assessed as damaged with *high confidence*; satellite imagery from late October 2023

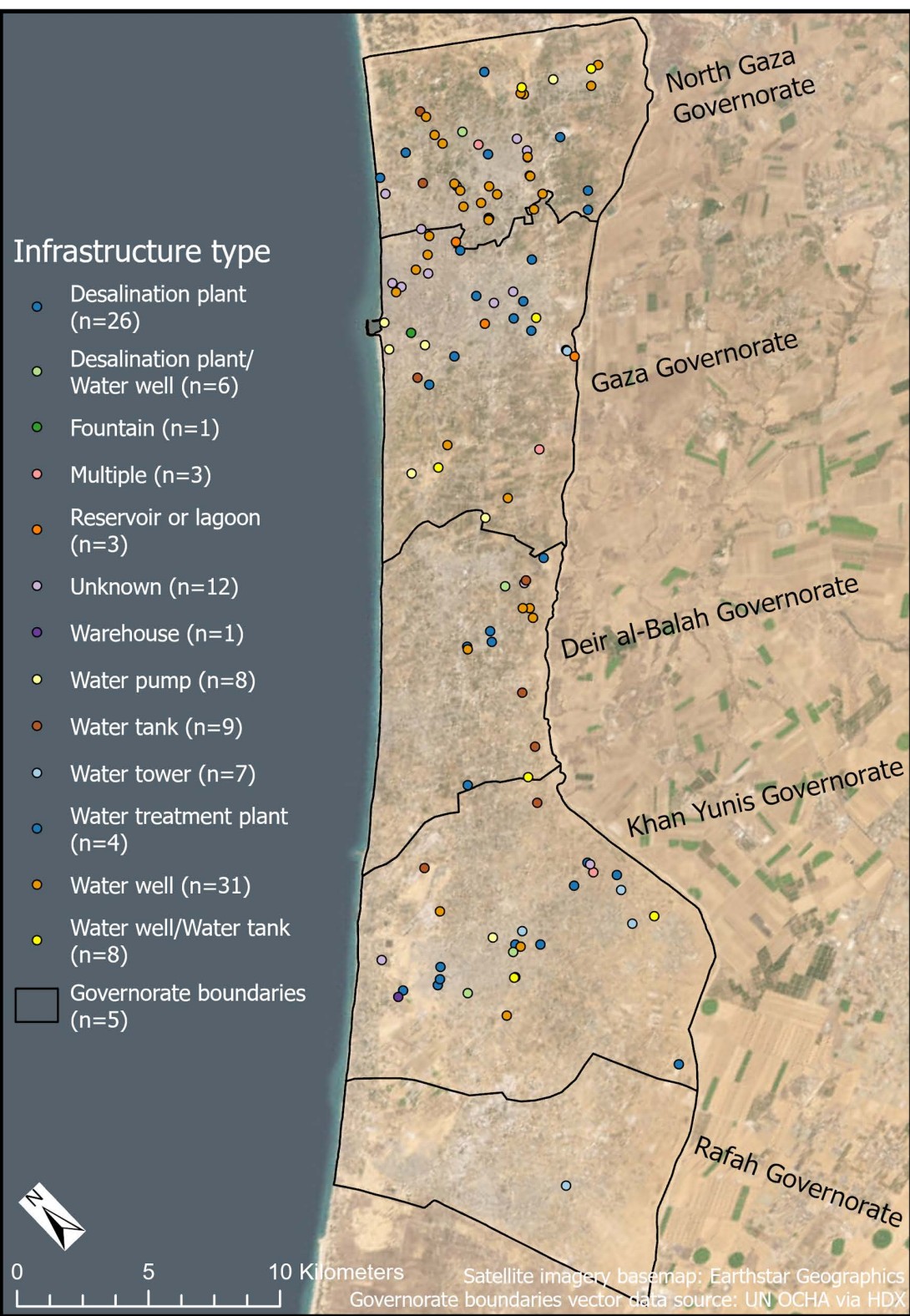

**Fig 5. Map of damaged WASH infrastructure sites by type.** Governorate boundaries vector data source: UN OCHA via Humanitarian Data Exchange. Satellite imagery basemap: Earthstar Geographics via ESRI.

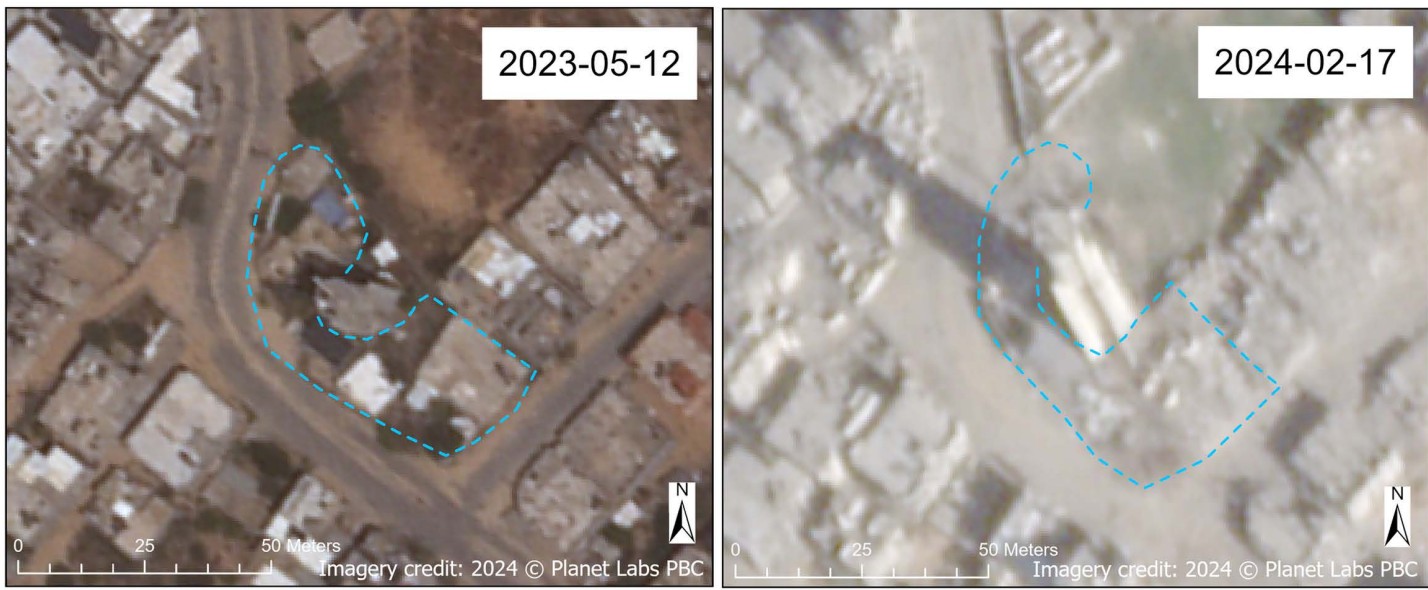

**Fig 6. Damaged Ayyah well desalination plant.** Before (May 2023) and after (February 2024) SkySat satellite images of the Ayyah Well Desalination Plant in Khan Yunis Governorate. Damaged areas of the facility are annotated in blue. Note that what appears to be a water storage tank, standing at a height of approximately 30 meters (roughly 100 feet) above ground level based on its shadow in the 2024 image, appears undamaged. (The annotations between the two images are different because of the steeper off-nadir angle in the 2024 image. The 2023 image is imaged closer to nadir. The sensor azimuth angles between the two images are comparable.) Imagery credit: Planet Labs PBC. Published under a CC BY license, with permission from Planet Labs PBC, original copyright 2024.

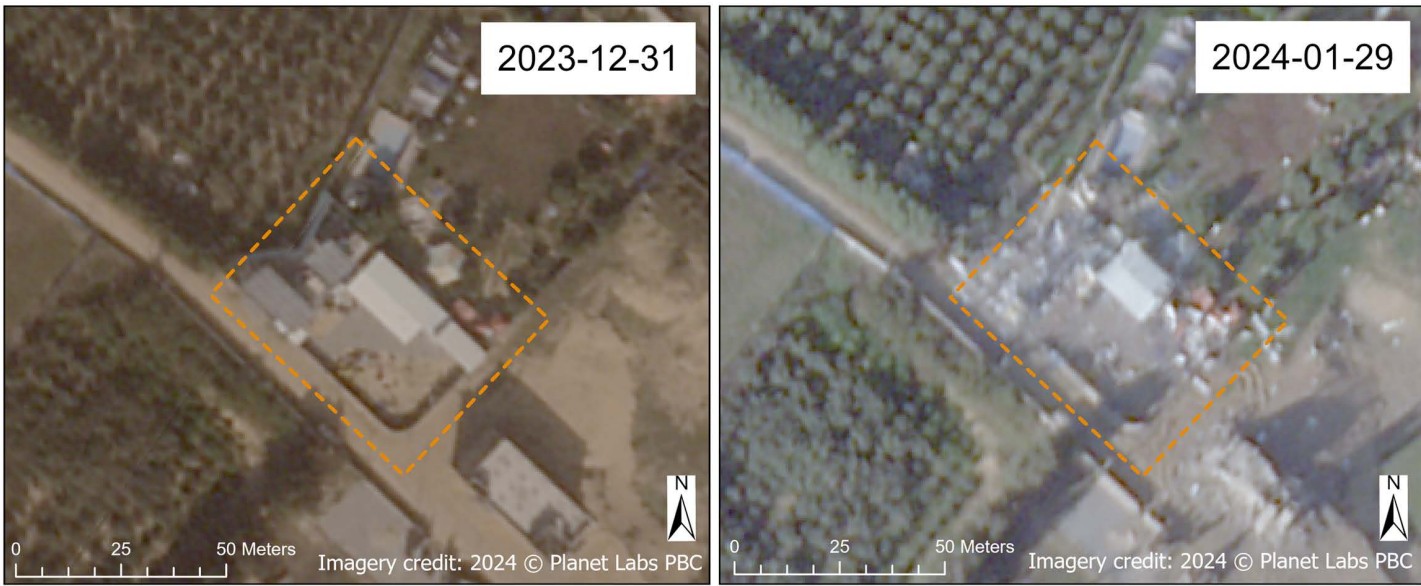

**Fig 7. Damaged Zahra sewage pumping station.** Before (December 2023) and after (January 2024) SkySat images of the Zahra sewage pumping station in Gaza Governorate. Damaged areas of the facility are annotated in orange. Imagery credit: Planet Labs PBC. Published under a CC BY license, with permission from Planet Labs PBC, original copyright 2024.

showed that the facility's solar panels were partially damaged but that by February 2024, the solar panels were completely destroyed. Further damage to the plant was noted in the February 2024 image, including to buildings that are believed to be part of the control infrastructure or administration of the plant (Fig 14).

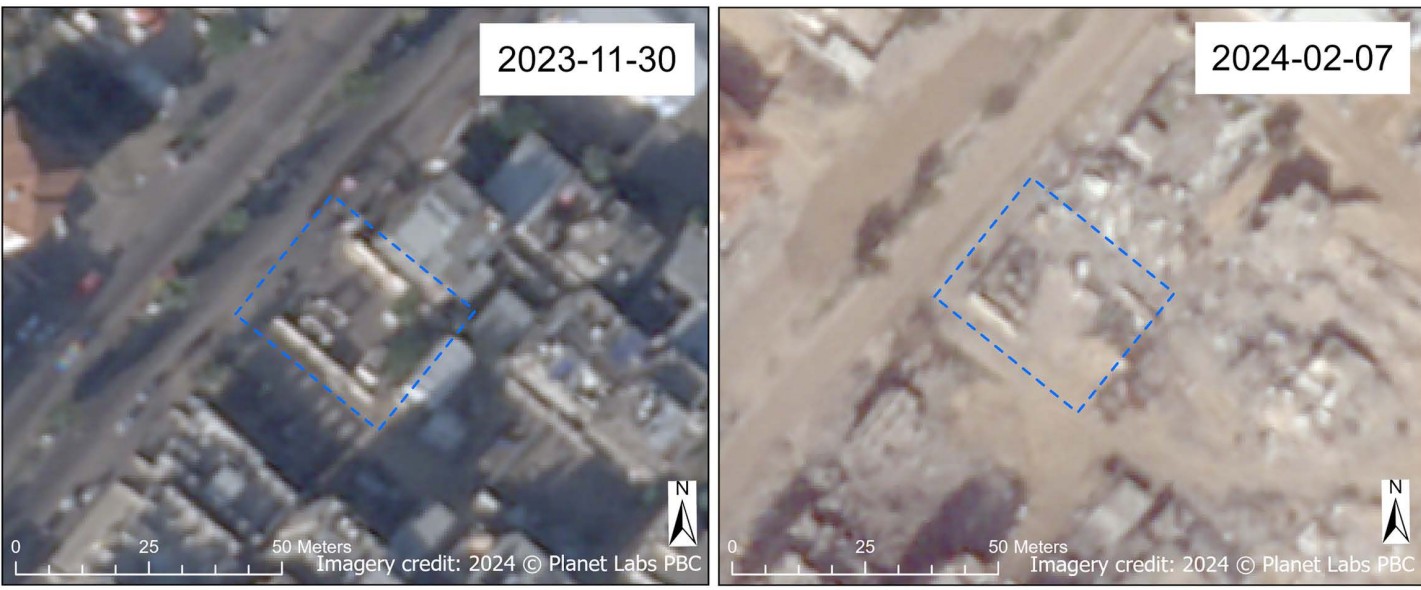

**Fig 8. Damaged Bani Suhaila sewage pumping station.** Before (November 2023) and after (February 2024) SkySat images of the Bani Suhaila sewage pumping station in Khan Yunis Governorate. Damaged areas of the facility are annotated in blue. Imagery credit: Planet Labs PBC. Published under a CC BY license, with permission from Planet Labs PBC, original copyright 2024.

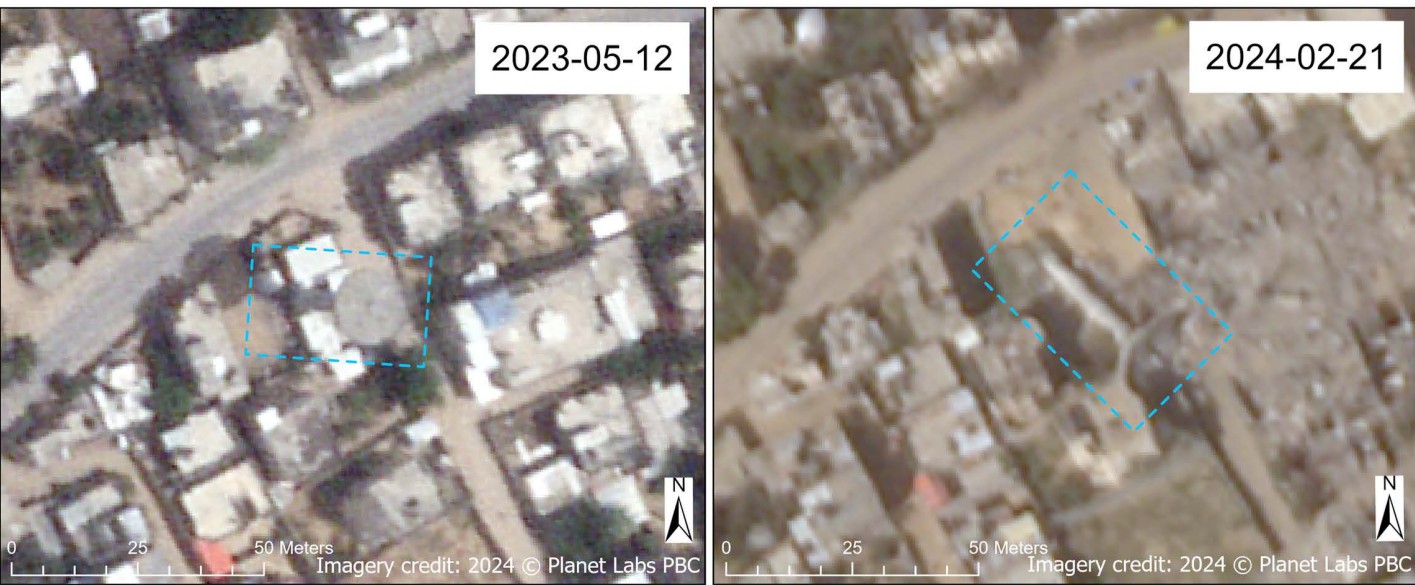

**Fig 9. Damaged water tower.** Before (May 2023) and after (February 2024) SkySat images of a water tower (annotated in blue) in Rafah Governorate that appears to have fallen on its side. Also note the visibly evident destruction of buildings, including the Umar bin Abd Al-Aziz Mosque, immediately to the east. Imagery credit: Planet Labs PBC. Published under a CC BY license, with permission from Planet Labs PBC, original copyright 2024.

Damage to (or the removal of) solar panels was also observed at other WASH infrastructure sites. Power generated by solar panels is often necessary for facility functionality and can provide additional means of energy generation; this is especially important in the Gaza Strip where fuel for generators is limited [66]. At the Khan Yunis Wastewater Treatment Plant

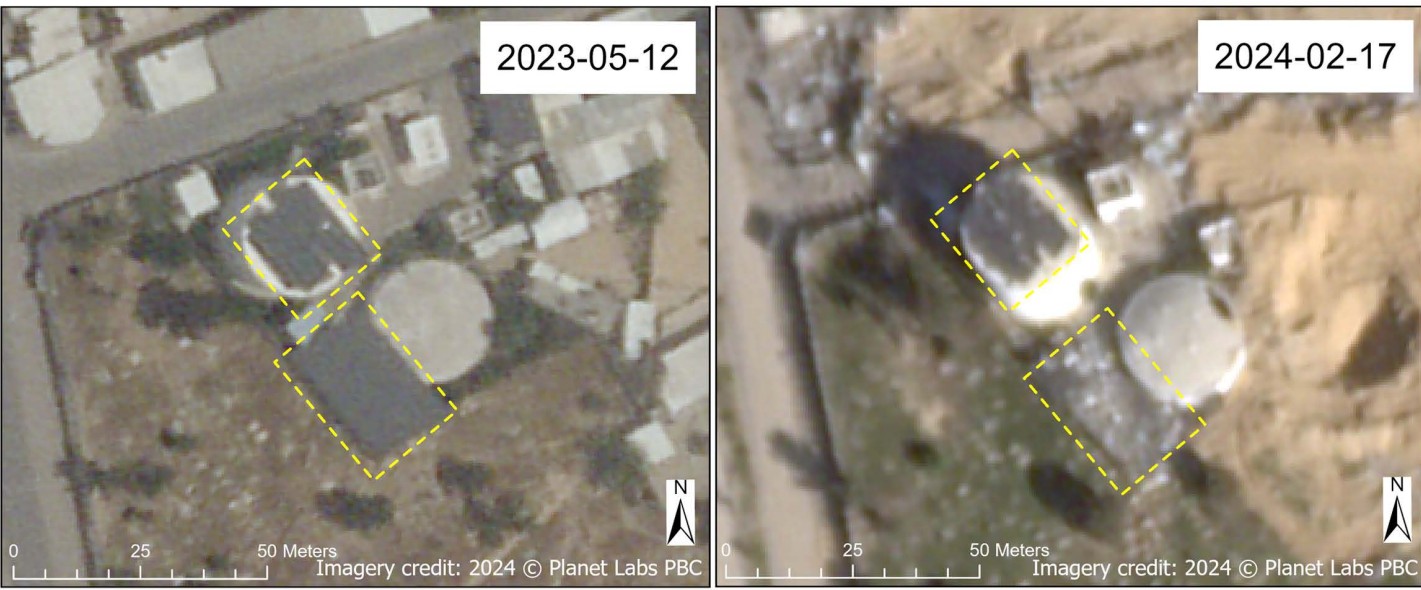

**Fig 10. Damaged Maan and Maan New water tanks and water wells.** Before (May 2023) and after (February 2024) SkySat images of Maan and Maan New water tanks and water wells in Khan Yunis Governorate. Damage to solar panels is annotated in yellow. Imagery credit: Planet Labs PBC. Published under a CC BY license, with permission from Planet Labs PBC, original copyright 2024.

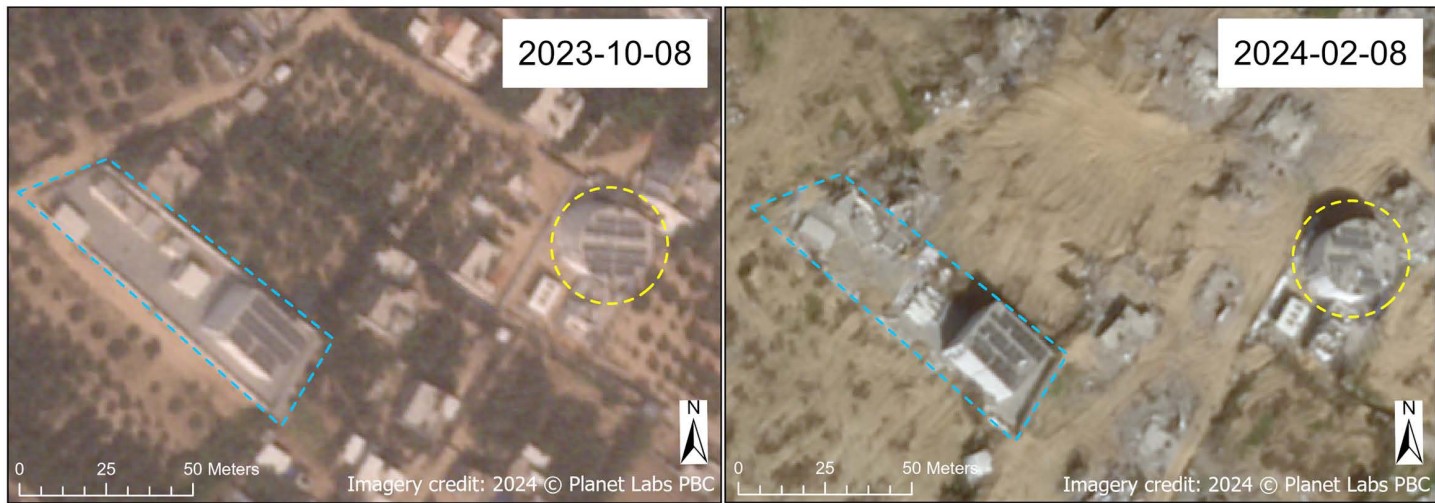

**Fig 11. Damaged BWSS (bulk water supply scheme) water tank.** Before (October 2023) and after (February 2024) SkySat images depicting damage to solar panels on top of a BWSS water tank (annotated in yellow) in Deir Al-Balah Governorate. Approximately 100 meters to the southwest is another damaged BWSS location (annotated in blue). Wide-area destruction is observed in the immediate vicinity. Imagery credit: Planet Labs PBC. Published under a CC BY license, with permission from Planet Labs PBC, original copyright 2024.

(Fig 15), deterioration of solar panel infrastructure over time was observed at the southwest corner of the plant; however, the cause of this degradation was not immediately clear based on imagery analysis alone. Fig 15 presents a four-date time series showing the deterioration of solar panel infrastructure at the site. Further investigation into this specific WASH site could elucidate whether deterioration of solar panel infrastructure observed here and in similar

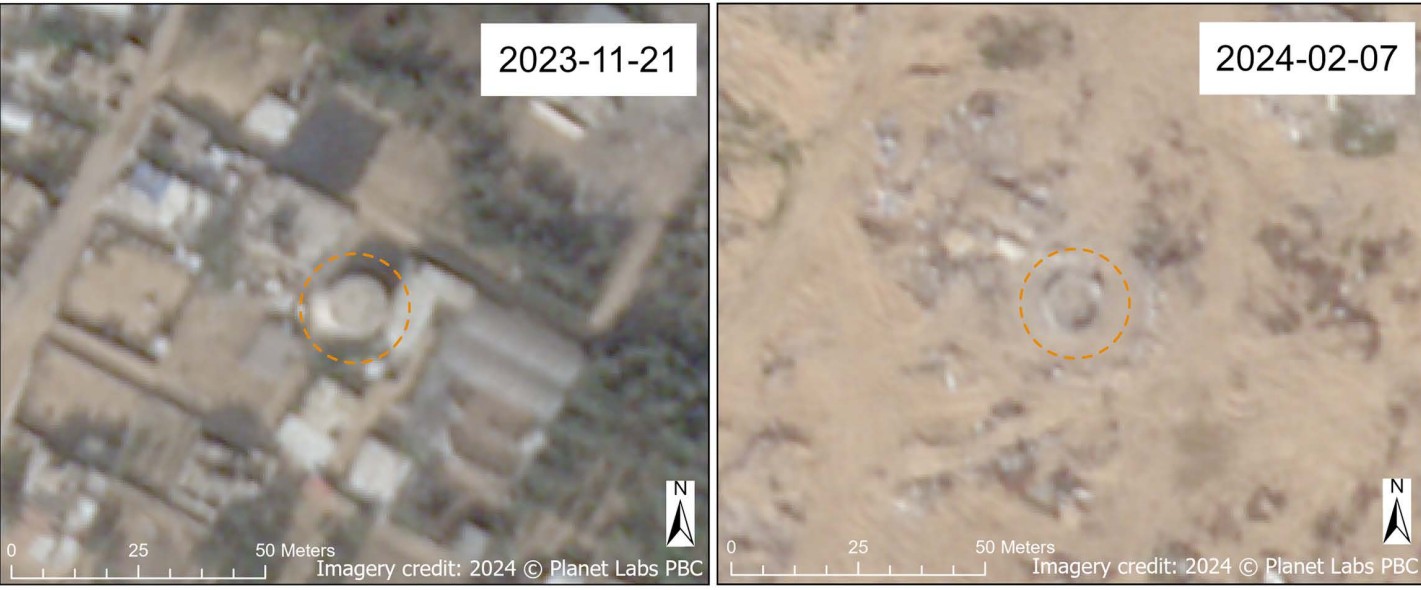

**Fig 12. Damaged Al Qarara BWSS water tank.** Before (November 2023) and after (February 2024) SkySat images of the Al Qarara BWSS water tank (also known as the "Qarara Old" water tank, according to CMWU) in Khan Yunis Governorate (annotated in orange). Note the widespread destruction of the area immediately surrounding the water tank in the 2024 image. Imagery credit: Planet Labs PBC. Published under a CC BY license, with permission from Planet Labs PBC, original copyright 2024.

cases was due to conflict-related damage or other factors, such as repurposing of functional solar panels.

## Discussion

### Relating WASH infrastructure damage to public health outcomes

Our determination that roughly half of WASH facilities in the Gaza Strip have been damaged since October 7, 2023, likely represents a conservative assessment of the actual damage during the study period. Although examining site-specific relationships between specific infrastructure damage and water-related health risks is beyond the scope of this study, the geographic breadth of WASH infrastructure damage, as well as the range of types of damaged infrastructure — from water treatment plants through wells and storage infrastructure — suggests significant public health impacts, including limited access to safe drinking water and exposure to wastewater. From October 2023 to May 2024, reports indicated water availability ranging from 1–8 liters per person per day [15,27,32], with the most severe reports indicating 1–3 liters per person per day [31–35]. Barely a week into the escalation of the conflict, UN OCHA reported that average water consumption had fallen to three liters per person per day [33]. The following month, Oxfam stated that water availability was between 1–3 liters per person per day [32], and UNICEF stated in December 2023 that recently displaced children in the southern Gaza Strip had access to only 1.5–2 liters per person per day [34]. A March 2024 Integrated Phase Classification (IPC) publication cited a February 2024 WASH Cluster study with a sample size of 1,200 people that estimated access to water was 1.5 liters per person per day [31]. Another WASH Cluster rapid assessment conducted the same month (February 2024) at 41 sites covering over 600,000 people found similar results, estimating the median accessible water volume to be 2 liters per person per day [31]. The Middle East Institute

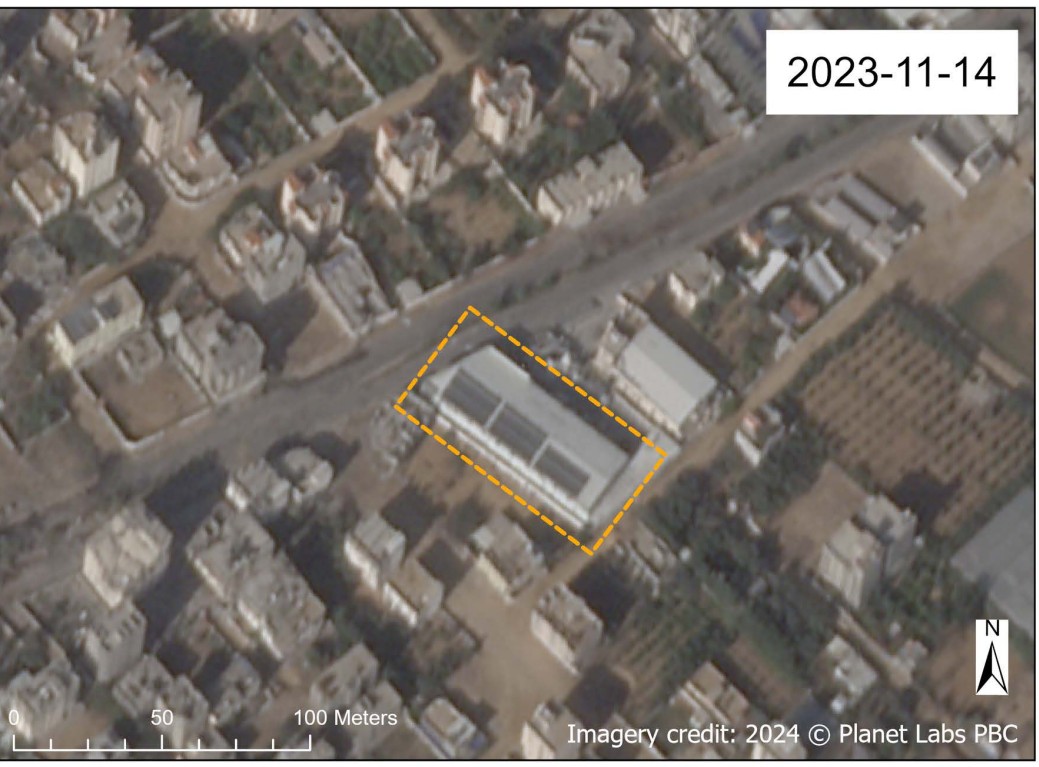

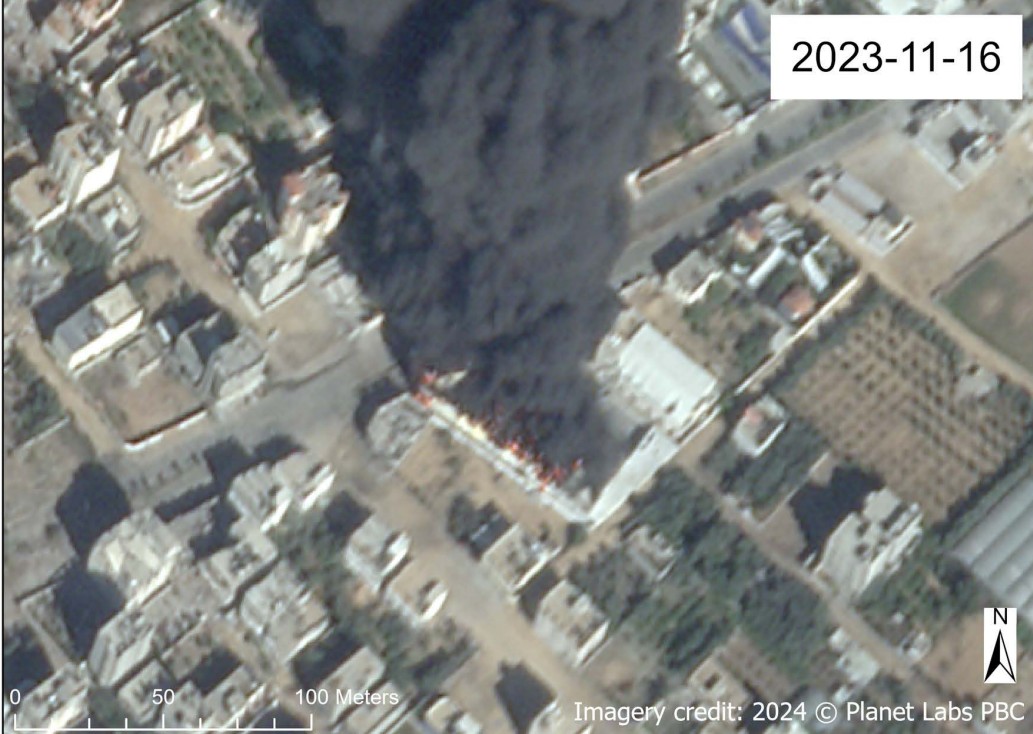

**Fig 13. Eta water plant on fire.** Before (November 14, 2023, top) and after (November 16, 2023, bottom) SkySat images of the Eta water plant (also known as the Abdul Salam Yaseen Company desalination plant) in Gaza Governorate. The desalination plant is annotated in orange in the before image. The desalination plant is showing flames and heavy black smoke in the after image captured on November 16, 2023, at 10:04:32 local time (08:04:32 UTC). Imagery credit: Planet Labs PBC. Published under a CC BY license, with permission from Planet Labs PBC, original copyright 2024.

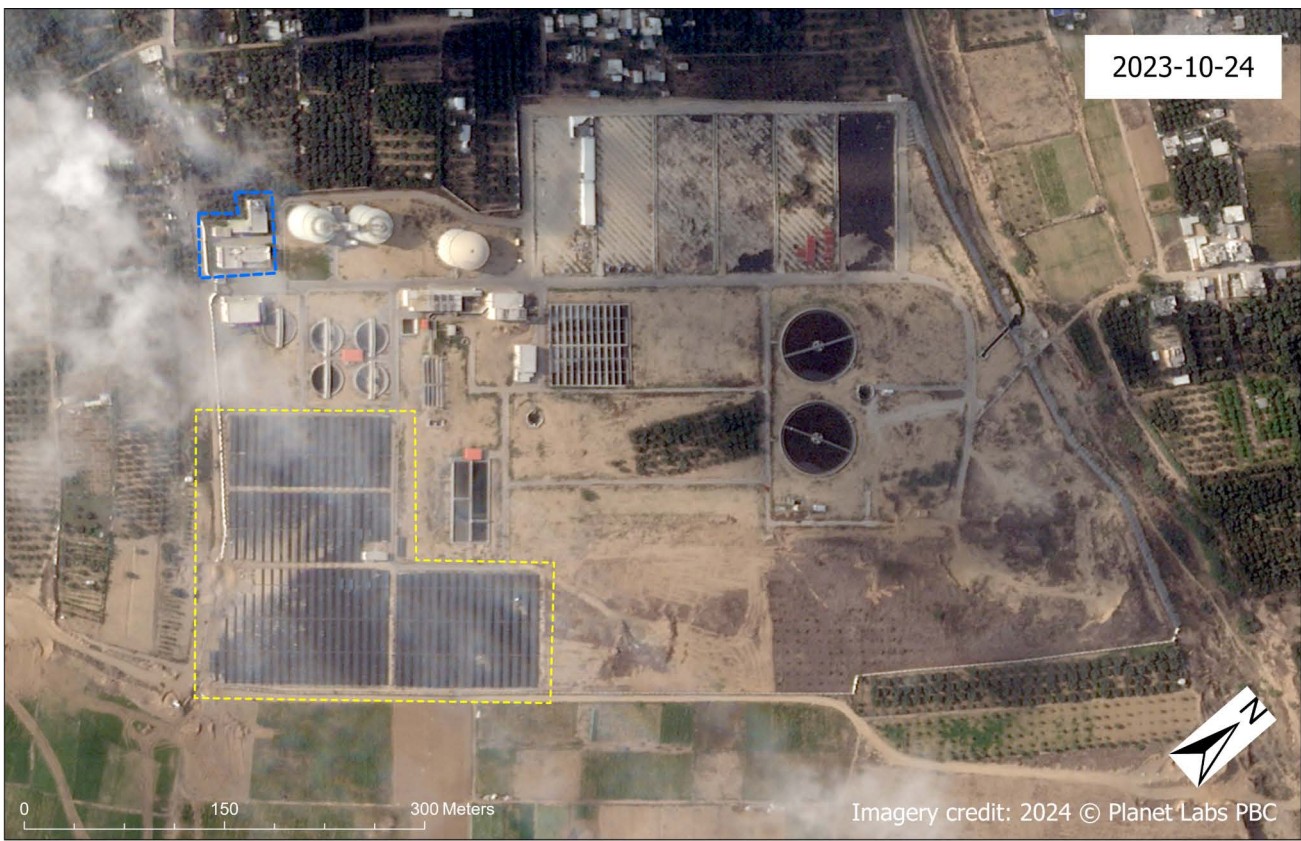

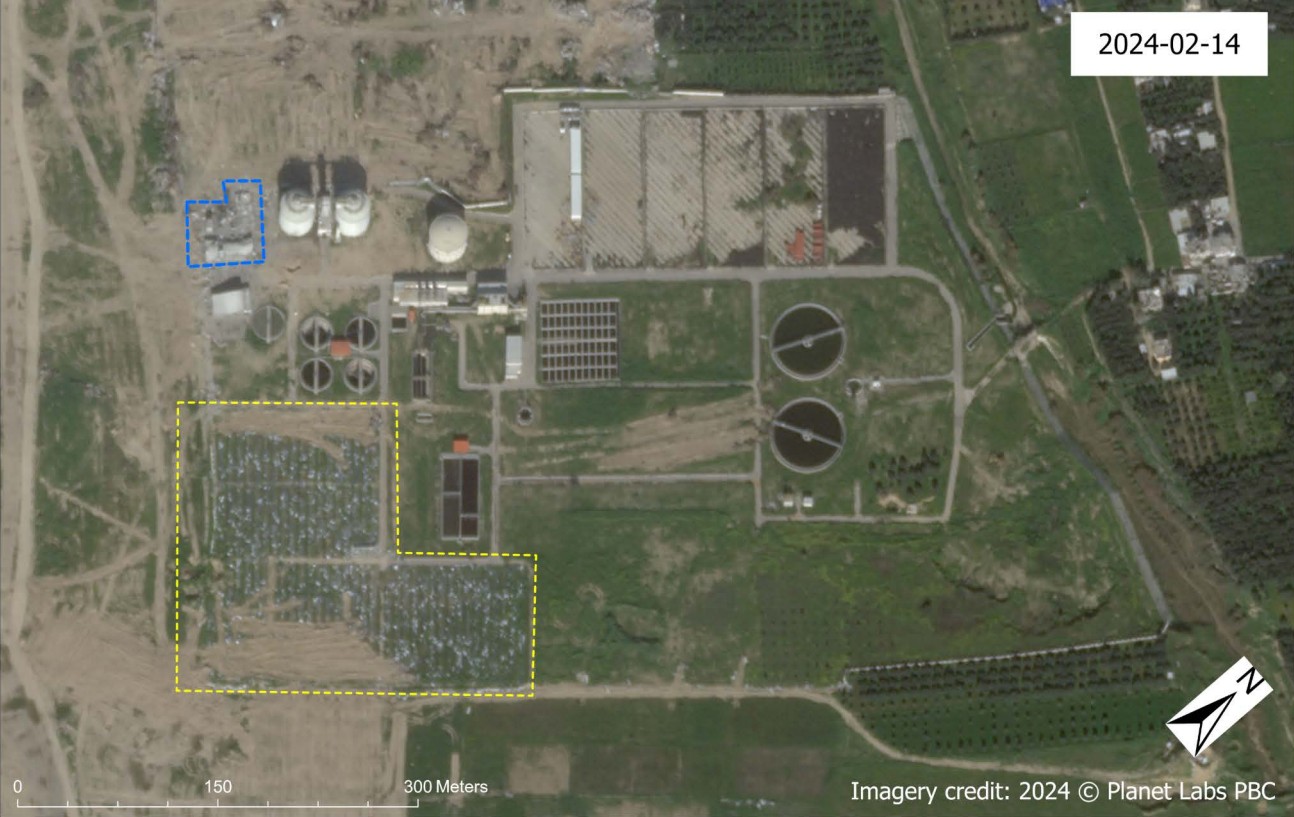

**Fig 14. Damaged Bureij Central Wastewater Treatment Plant.** Before (October 2023, top) and after (February 2024, bottom) SkySat images of the Bureij Central Wastewater Treatment Plant in Deir Al-Balah Governorate. In the before image, note partial damage to solar panels (annotated in

yellow) on the left side of the image. Note complete destruction of solar panels in the after image. The blue annotation represents buildings that are believed to be part of the control infrastructure or administration of the plant that are damaged in the after image. Imagery credit: Planet Labs PBC. Published under a CC BY license, with permission from Planet Labs PBC, original copyright 2024.

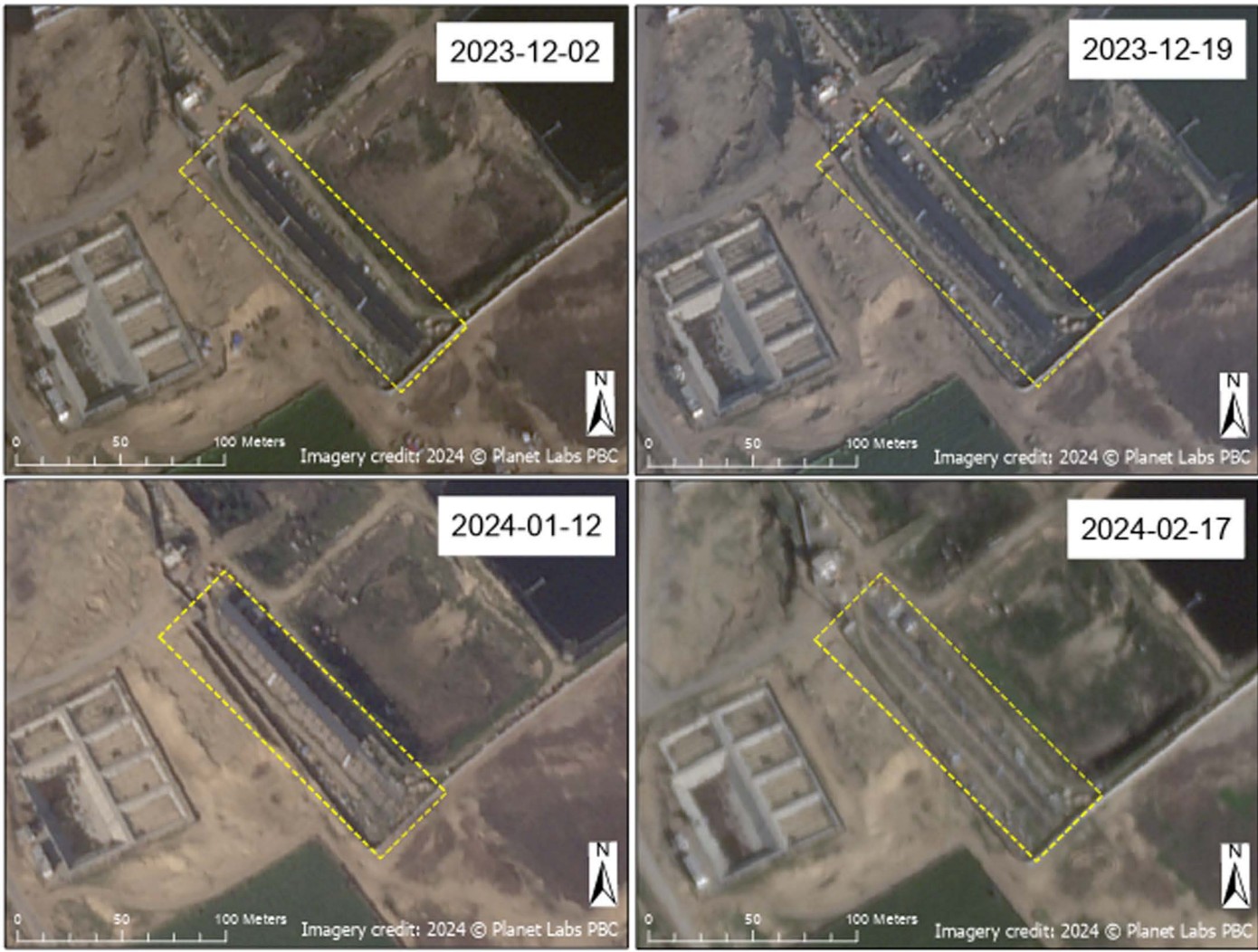

**Fig 15. Solar panel deterioration.** Time series of SkySat images showing the deterioration of solar panel infrastructure (annotated in yellow) located at the Khan Yunis Wastewater Treatment Plant. Imagery credit: Planet Labs PBC. Published under a CC BY license, with permission from Planet Labs PBC, original copyright 2024.

estimated in March 2024 that water availability was between 1.5–3 liters per person per day [35]; a June 2024 UNEP report stated that in April 2024, water availability was estimated to be between 2–8 liters per person per day [15]; and a July 2024 Oxfam report stated that their own analysis of water sources data available in WASH Cluster meeting notes covering 208 days from November 1, 2023, to May 26, 2024, concluded that water availability was on average 4.74 liters per person per day [27]. These findings suggest that water availability (see S1 Appendix for further details) has remained critically low throughout the conflict, likely with severe implications for public health.

There have been myriad anecdotal reports of Palestinians in Gaza going to extreme lengths to cope with water insecurity (which has been defined as "having insufficient access to safe water for consumption and use" [67]) due to damage to and destruction of WASH infrastructure. Consumption of brackish water extracted from agricultural wells was documented as early as mid-October 2023 [33]. An internally-displaced person sheltering at a school in Khan Yunis said in November 2023: "Most of the water in the [Gaza Strip] is polluted. People in the shelters drink polluted water, there is no potable water and it's not provided by UNRWA" [68]. By March 12, 2024, the Gazan Ministry of Health reported that 27 people had died of malnutrition and dehydration at hospitals in northern Gaza [69]. Of the "1,091 families displaced by Israel's evacuation orders in Rafah in May [2024] who stayed at NRC-supported sites," the Norwegian Refugee Council reported that "57% had no access to safe water" [70]. Emphasizing the grim WASH situation, WHO representative for the occupied Palestinian Territory Rik Peeperkorn said in June 2024 that "[w]e all know that malnutrition is not just dependent on food. You need the right food and the right food groups, etc. But of course, it links as well to the water and sanitation situation, which is dire everywhere…" [71].

Wastewater leaching into the environment due to damage to or non-operation (from fuel shortages) of wastewater infrastructure can also have dangerous health effects and has been documented in the current conflict [14,15,58,59]. Damage to or disruption of wastewater treatment plants "leads to the release of untreated sewage into the environment, contaminating beaches and coastal waters, soils and potentially the groundwater. Untreated sewage contains pathogens, nutrients, particulate organic matter, plastics and hazardous chemicals. The presence of sewage in the environment poses an immediate threat to human health through direct exposure to pathogens" [15]. In March 2024, an estimated 60,000 $m^3$/day of wastewater and sewage was discharged into the environment, primarily into the Mediterranean Sea [15]. Previous anomalous wastewater discharges in the Gaza Strip as a result of past conflicts led to an increase in the number of cases of diarrhea (especially in children) and the contamination of groundwater and agricultural land with heavy metals [15].

Damage to WASH infrastructure may also create negative feedback loops. For instance, when a water pipeline leaks, civilians may resort to informal water sources, reducing public utility usage and revenue. This, combined with the loss of leaking water that will not result in revenue, hampers the utility's ability to repair the leak, perpetuating the cycle [21]. External financial support from international groups or governmental agencies can help break this loop. WASH infrastructure repair may also be hampered by security concerns or access restrictions, a lack of available materials (or restrictions on "dual-use" materials), and insufficient funds. The lack of fuel availability also has multiple knock-on effects, including the inability to power treatment plants or for individuals to boil water for safe consumption [21,66].

While difficult to link damage of specific WASH infrastructure to individual- or population-level disease outcomes, evidence from other armed conflicts in Syria, Yemen, Ethiopia, and Ukraine shows a clear link between damage to WASH infrastructure and adverse public health outcomes. For instance, between 2011 and 2019, researchers noted an increase in diarrheal disease in Syria's Idlib and Aleppo Governorates, coinciding with attacks on WASH infrastructure by both state and non-state actors [72]. Although a direct causal link could not be established due to confounding variables, the lack of clean water created an ideal environment for the spread of communicable diseases, such as polio, hepatitis A, and diarrheal diseases such as cholera [72,73]. Polio re-emerged in Syria in 2013, the same year that a lack of water meant that individuals were insufficiently decontaminated following a chemical weapons attack in Damascus [72]. Similarly, armed conflict in Yemen degraded an already fragile WASH infrastructure system, resulting in its collapse and a cholera outbreak [74]. In Ethiopia,

conflict-related damage to more than half of the water supply systems in the Tigray region resulted in drastically reduced access to clean water, exposing millions of people to water-related health risks [24]. And in Mariupol, Ukraine, over 40% of the water supply system was damaged by conflict, raising concerns from the WHO in May 2022 of a potential cholera outbreak due to the contamination of drinking water with sewage [25]. Taken together, these case studies illustrate the devastating public health risks posed by conflict-related WASH infrastructure damage. While the effects of such damage vary based on local contexts, these examples highlight the need for future research to explore causal relationships between WASH infrastructure attacks and public health outcomes.

## Comparing our results with previously published results

In the absence of field data on WASH infrastructure damage, we gauged our agreement with several WASH infrastructure-focused analyses. Our finding of 49.8% damage aligns closely with other published assessments and is within approximately 7.2% of all known similar analyses conducted using similar study periods (detailed in S2 Appendix). These include: a January 2024 World Bank/Ipsos analysis, which found approximately 57% of WASH facilities had been damaged or destroyed [47]; a January 2024 UNOSAT analysis, which found an average of 55% of WASH facilities damaged or destroyed based on assessments from the five governorates [47]; a February 2024 Global WASH Cluster analysis, which found that 56% of water facility points and 56% of sewage facility points had been damaged [31]; and a May 2024 BBC analysis, which found that 53% of water facilities had been damaged or destroyed [49]. Taken together, results from these disparate sources, including our analysis, suggest that roughly half of the WASH infrastructure in the Gaza Strip had been damaged by early 2024.

## Protection of water infrastructure in international law

Empirical research on WASH infrastructure in the Gaza Strip and other conflict contexts is needed to provide basic facts as to the condition of WASH infrastructure for potential investigation by intergovernmental organizations and justice and accountability mechanisms and to inform political debates around enforcing the law of armed conflict. International Humanitarian Law (IHL) "is often violated, and the political will required to enforce it remains largely theoretical. The challenge to those keen on reducing harm to civilians and ensuring water for peace and prosperity thus becomes to provide evidence to support the legal and political frameworks, and to inform military doctrine and training manuals" [75]. The establishment of "causal chains" from armed conflict damage to undesirable health outcomes (e.g., increased prevalence of waterborne diseases) is key to understanding war's effects on public health.

In May 2024, the Prosecutor of the International Criminal Court, Karim Khan, announced that he had filed applications for arrest warrants for Israeli Prime Minister Benjamin Netanyahu and then-Israeli Defense Minister Yoav Gallant for alleged war crimes and crimes against humanity for their conduct with respect to civilians in the Gaza Strip since October 7, 2023 [76]. Specifically, the OTP (Office of the Prosecutor) stated that evidence that his office had collected "shows that Israel has intentionally and systematically deprived the civilian population in all parts of Gaza of objects indispensable to human survival," including water. The OTP alleged that the "total siege over Gaza [...] included cutting off cross-border water pipelines from Israel to Gaza – Gazans' principal source of clean water – for a prolonged period beginning 9 October 2023, and cutting off and hindering electricity supplies from at least 8 October 2023 until today [May 20, 2024]" [76]. (The Israeli government's usage of the phrase "complete siege" has been reported in media articles, including October 2023 articles (on October 9 and October 10) from The New York Times in which Gallant was quoted saying

that "no electricity, no food, no water, no fuel" would be allowed to enter the Gaza Strip [77,78].) The OTP alleged that these and other tactics (including dehydration) were used as a form of collective punishment of Gazan civilians, war conduct which is prohibited by International Humanitarian Law [76].

Moreover, in a contemporaneously published report by a panel of experts in international law convened by the OTP, the experts unanimously concluded that "the acts through which this war crime was committed include a siege on the Gaza Strip and [...] cutting off supplies of electricity and water, and severely restricting food, medicine and fuel supplies. This deprivation of objects indispensable to civilians' survival took place in the context of attacks on facilities that produce food and clean water..." [79]. The panel clarified that water is considered indispensable for the survival of civilians, writing that the "war crime of 'intentionally using starvation of civilians as a method of warfare' requires 'depriving [civilians] of objects indispensable to their survival,'" and that "[t]he crime is not limited solely to the deprivation of food, but includes other objects indispensable for the survival of civilians such as water, fuel and medicine" [79].

Evidence showing the intentional deprivation of access to water — whether through siege tactics or destruction of WASH infrastructure — could help legal practitioners and accountability mechanisms determine whether violations of International Humanitarian Law and International Criminal Law have occurred. Empirical research showing the extent and severity of WASH infrastructure damage combined with other sources of evidence could aid in determinations of whether destruction of WASH infrastructure — deemed to be objects indispensable to civilian survival — was systematic and widespread. Future legal proceedings could utilize satellite-based remote sensing methods, such as those used in this study, to help build a corpus of evidence that would naturally include other sources of "traditional" legal evidence, such as witness testimony, physical evidence, and government documents.

## Limitations and opportunities for future study

This study had several unavoidable limitations due to the remote, satellite image-based analytical approach and challenges posed by active conflict, which severely curtails field-based assessment of the relationships between WASH infrastructure damage and water-related health risks. Satellite imagery can detect physical damage, but it does not account for factors like lack of fuel, parts, funds, or personnel that could render a facility non-functional even if it appears undamaged. We also acknowledge that there might have been WASH infrastructure sites with damage that we labeled as *no damage visible*, given the limitations of the sensor on Planet's SkySat satellites, as well as the fact that most of our source data was point vector data, not polygon vector data, which would have clearly delineated WASH site boundaries. Satellite imagery provides a means of analyzing damage to WASH infrastructure while gaining physical access on the ground may be dangerous, difficult, or impossible. While remote sensing analysis can provide important, up-to-date facts about the physical condition of WASH infrastructure, it cannot provide comprehensive information as to the operating status of the facilities themselves. As a result, we could not determine, for example, whether a given site was still functional despite an apparent lack of evident damage. Future research would benefit from ground-truth assessments by experts who can gain physical access to WASH sites and consider public health impacts to localized damage, even if done in a retrospective fashion after the cessation of the conflict.

An additional limitation results from not having a comprehensive dataset of all WASH infrastructure locations and types in the Gaza Strip. Without a complete pre-conflict dataset, we will always underestimate the level of damage to infrastructure, regardless of analytical approach. Although the rapidity with which WASH infrastructure changes (due to the

importance of water for survival) obviates researchers' ability to obtain a fully comprehensive and accurate dataset at any given time, future studies should seek to incorporate additional data on the pre-October 7 WASH infrastructure locations using data sources not considered here. Despite our efforts to collate as much verifiable data as possible, we acknowledge that our dataset is inherently incomplete and may contain errors or inaccuracies, even though a good faith and systematic effort was made to verify each point. We focused on non-residential, above-ground water infrastructure visible in satellite imagery; we did not include data on residential water storage tanks, subterranean infrastructure, cross-border pipelines, water transfer networks within the Gaza Strip, or agricultural ponds. Notably, our findings on water well damage are likely an underestimate due to the exclusion of locations where sufficient pre-conflict data were unavailable. This strict inclusion criterion almost certainly removed legitimate wells from our analysis for the sake of higher confidence. Water wells, difficult to detect via satellite imagery alone, could be better captured in future studies with expanded data.

Understanding trends or patterns of damage to specific types of WASH infrastructure is valuable because it could shed light as to how these sites are being damaged: are they being intentionally targeted because of their criticality to water supplies, or is damage to WASH infrastructure so dispersed geographically and by type as to be random? Future studies should conduct statistical analyses to test, for example, if the physical size of a WASH facility made it more or less likely to be damaged or if there are any other inferences that can be drawn based on facility type.

Finally, we know that a large proportion of the population in the Gaza Strip has been repeatedly internally-displaced during the conflict. Our analysis does not account for pre-conflict or mid-conflict population distributions in the present study and cannot convey if undamaged WASH infrastructure in various locations provided adequate access to safe water or how concentrations of displaced populations aligned with damaged WASH infrastructure hotspots, for example. Future studies should explore how population shifts exacerbate water-related risks in conflict settings.

## Conclusion

Our remote sensing analysis of damage to WASH infrastructure in the Gaza Strip between October 8, 2023, and February 22, 2024, concluded that roughly half (49.8%, n = 119) of WASH infrastructure locations in our dataset for which very-high resolution optical satellite imagery was available and interpretable (n = 239) had sustained conflict-related damage. Damage to WASH infrastructure is particularly deleterious to the survival and well-being of a population, given the long-lasting and wide-ranging second- and third-order effects that damage to water systems can induce. Damage to WASH infrastructure in the Gaza Strip since the escalation of hostilities on October 7, 2023, has been "found to be one of the factors which have led to the deaths of dozens of new-borns due to dehydration, hundreds of thousands of cases of acute diarrhoea, scabies and acute hepatitis-A" [80]. Experts from the Geneva Water Hub (GWH) have warned that introduction of cholera would pose potential devastating health outcomes, emphasizing the importance of understanding how damage to WASH infrastructure affects civilians in conflict zones [80]. The GWH experts have called for a detailed examination of "the evidence observed and cited in this study." Our intent with this research is to meet that need, even if only to a limited extent, and to fill any knowledge gaps caused by limited studies on the topic, lack of available data, or insufficient transparency.

Water is critical to survival, and WASH infrastructure must be protected by all parties to any conflict to ensure access to clean water by local populations, as required by international law. Our study contributes to the body of knowledge regarding critical civilian infrastructure in the current conflict in Gaza in that we documented — in a systematic and academically

rigorous manner — damage to many key pieces of WASH infrastructure. Results from this study provide supporting empirical data and reinforce assertions by other researchers that the capacity of water systems in the Gaza Strip since the escalation of hostilities on October 7, 2023, has been curtailed and therefore produced negative health results (up to and including death) for Palestinians living in Gaza.

The reconstruction needs are immense; even if water treatment and production capacity improves within Gaza or water transfers from Israel return to pre-October 7 levels, the distribution systems remain inadequate (e.g., lack of fuel for water trucks; damage to water transfer pipeline infrastructure). Estimated costs of WASH infrastructure rehabilitation or reconstruction in the Gaza Strip are in the several hundred million US Dollars range. For example, an unpublished May 2024 CMWU report estimated the reconstruction/recovery cost at $310,500,000, while in a joint March 2024 interim damage assessment report with the EU and UN, the World Bank estimated that damage to WASH infrastructure amounted to $503,000,000 as of January 2024 [48].

Results from this paper provide critical empirical evidence that can inform conversation and policy for reconstruction of WASH infrastructure in the Gaza Strip once a ceasefire agreement is enacted and the development process can begin in earnest — a process that is expected to take many years. In addition to informing development professionals, it can also give researchers, humanitarians, and governmental officials a solid empirical knowledge base to inform their work of providing life-saving aid to those in an active conflict zone. We posit that this research has far-reaching impacts and goes beyond work related to the Gaza Strip; indeed a similar WASH infrastructure damage assessment could be applied in other conflict contexts around the world, such as in Ukraine, Sudan, Yemen, and further afield in future conflict contexts currently non-existent.

## Supporting information

**S1 Appendix.  Water availability reports in the Gaza Strip.**
(DOCX)

**S2 Appendix.  Third-party organization WASH damage assessments [81,82].**
(DOCX)

## Acknowledgments

The authors wish to acknowledge and thank all members of the Decentralized Damage Mapping Group (DDMG), which is an inter-institutional research group focused on remote sensing methods for conflict damage assessment, especially the following DDMG members: Corey Scher, Wim Zwijnenburg, Marie Schellens, Lina Eklund, and Rob Watson. We also wish to thank the following individuals for informing our understanding of the current conflict context as it pertains to water systems and health-related effects: Zeina Jamaluddine, Rich Pauloo, Juliane Schillinger, Amir Mohareb, Sera Young, and Fiona Majorin. We wish to thank Jan Selby of the University of Leeds, who provided valuable background information adding context to the research presented in this study.

## Author contributions

**Conceptualization:** Brian Perlman, Jamon Van Den Hoek.

**Data curation:** Brian Perlman.

**Formal analysis:** Brian Perlman.

**Investigation:** Brian Perlman.

**Methodology:** Brian Perlman, Jamon Van Den Hoek.

**Project administration:** Jamon Van Den Hoek.

**Supervision:** Jamon Van Den Hoek.

**Validation:** Brian Perlman.

**Visualization:** Brian Perlman.

**Writing – original draft:** Brian Perlman, Shalean M. Collins.

**Writing – review & editing:** Shalean M. Collins, Jamon Van Den Hoek.

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
