## [Decision Letter · Decision Letter 0]

7 Nov 2024

PGPH-D-24-02192

Public health implications of satellite-detected widespread damage to WASH infrastructure in the Gaza Strip

Dear Dr. Perlman,

Thank you for submitting your manuscript to PLOS Global Public Health. After careful consideration, we feel that it has merit but does not fully meet PLOS Global Public Health’s publication criteria as it currently stands. Therefore, we invite you to submit a revised version of the manuscript that addresses the points raised during the review process.

We look forward to receiving your revised manuscript.

Kind regards,

Nancy Angeline Gnanaselvam

Academic Editor

Journal Requirements:

2. We do not publish any copyright or trademark symbols that usually accompany proprietary names, eg (R), (C), or TM  (e.g. next to drug or reagent names). Please remove all instances of trademark/copyright symbols throughout the text, including © on pages 11, 12, 13 and 14.

3. Figures 3 to 15: please (a) provide a direct link to the base layer of the map (i.e., the country or region border shape) and ensure this is also included in the figure legend; and (b) provide a link to the terms of use / license information for the base layer image or shapefile. We cannot publish proprietary or copyrighted maps (e.g. Google Maps, Mapquest) and the terms of use for your map base layer must be compatible with our CC-BY 4.0 license. 

Additional Editor Comments (if provided):

Well written article which is of relevance to current geopolitical situation. Ethical approval and ethics of obtaining the data and conduct of the analysis need to be mentioned.

Reviewers' comments:

Reviewer's Responses to Questions

**Comments to the Author**

1. Does this manuscript meet PLOS Global Public Health’s publication criteria ? Is the manuscript technically sound, and do the data support the conclusions? The manuscript must describe methodologically and ethically rigorous research with conclusions that are appropriately drawn based on the data presented.

Reviewer #1: Yes

Reviewer #2: Yes

2. Has the statistical analysis been performed appropriately and rigorously?

Reviewer #1: Yes

Reviewer #2: N/A

3. Have the authors made all data underlying the findings in their manuscript fully available (please refer to the Data Availability Statement at the start of the manuscript PDF file)?

Reviewer #1: Yes

Reviewer #2: Yes

4. Is the manuscript presented in an intelligible fashion and written in standard English?

Reviewer #1: Yes

Reviewer #2: Yes

5. Review Comments to the Author

Reviewer #1: The abstract starts with '..Hamas terrorist attacks of October 7, 2023' and I wonder about the relevance of that statement in the overall objective of the paper. Nowhere else in the paper is a history of the conflict or the events leading up to the point discussed in detail, so I wonder why this statement is required in the abstract. In line with what the rest of the paper explores [and does so very well], the statement could just be 'Israel's violence in Gaza in the current offensive' or something on those lines. This statement infers the blame on Hamas - a point of debate which I'm sure the authors are not intending for the paper to explore.

Given the nature of the well documented destruction, it would e appropriate in the discussion to add the current international law regarding destruction of water related infrastructure. Amnesty has reports etc. I believe this will add more context to the destruction that has been documented in the paper.

Reviewer #2: Thank you for the opportunity to review this interesting manuscript. This study evaluates the location and extent of WASH facility damage in GAJA by using publicly available satellite images. I believe that the methods used in this study shed light on new approaches available to study public health issues in the 21st century. I have only a few minor comments.

In the discussion, I suggest adding 1-2 paragraphs with detailed examples from past conflict zones on the destruction of WASH facilities and subsequent health impacts. This would help to emphasize the public health significance of the study’s findings.

It would also be helpful to provide specific details on how to access the data (e.g., URL, retrieval methods) in the methodology section to facilitate replication by other researchers. Additionally, if any software was used to process and analyze the images, please specify it.

In Figure 4, using completely different colors for the legend might improve readability.

6. PLOS authors have the option to publish the peer review history of their article (what does this mean? ). If published, this will include your full peer review and any attached files.

**Do you want your identity to be public for this peer review?** For information about this choice, including consent withdrawal, please see our Privacy Policy .

Reviewer #1: No

Reviewer #2: No

---

## [Decision Letter · Decision Letter 1]

11 Dec 2024

PGPH-D-24-02192R1

Public health implications of satellite-detected widespread damage to WASH infrastructure in the Gaza Strip

Dear Dr. Perlman,

Thank you for submitting your manuscript to PLOS Global Public Health. After careful consideration, we feel that it has merit but does not fully meet PLOS Global Public Health’s publication criteria as it currently stands. Therefore, we invite you to submit a revised version of the manuscript that addresses the points raised during the review process.

We look forward to receiving your revised manuscript.

Kind regards,

Nancy Angeline Gnanaselvam

Academic Editor

Journal Requirements:

Editor Comments :

  Please, specify under the OHRP 2018 requirements, which category of exemption from IRB Review does this study fall under. A communication from the IRB should be ideally obtained regarding the same.

Reviewers' comments:

Reviewer's Responses to Questions

**Comments to the Author**

1. If the authors have adequately addressed your comments raised in a previous round of review and you feel that this manuscript is now acceptable for publication, you may indicate that here to bypass the “Comments to the Author” section, enter your conflict of interest statement in the “Confidential to Editor” section, and submit your "Accept" recommendation.

Reviewer #1: All comments have been addressed

Reviewer #2: All comments have been addressed

2. Does this manuscript meet PLOS Global Public Health’s publication criteria ? Is the manuscript technically sound, and do the data support the conclusions? The manuscript must describe methodologically and ethically rigorous research with conclusions that are appropriately drawn based on the data presented.

Reviewer #1: Yes

Reviewer #2: Partly

3. Has the statistical analysis been performed appropriately and rigorously?

Reviewer #1: Yes

Reviewer #2: N/A

4. Have the authors made all data underlying the findings in their manuscript fully available (please refer to the Data Availability Statement at the start of the manuscript PDF file)?

Reviewer #1: Yes

Reviewer #2: Yes

5. Is the manuscript presented in an intelligible fashion and written in standard English?

Reviewer #1: Yes

Reviewer #2: Yes

6. Review Comments to the Author

Reviewer #1: (No Response)

Reviewer #2: (No Response)

7. PLOS authors have the option to publish the peer review history of their article (what does this mean? ). If published, this will include your full peer review and any attached files.

**Do you want your identity to be public for this peer review?** For information about this choice, including consent withdrawal, please see our Privacy Policy .

Reviewer #1: **Yes: ** Harsh

Reviewer #2: No

---

## [Editor Report · Decision Letter 2]

9 Jan 2025

Public health implications of satellite-detected widespread damage to WASH infrastructure in the Gaza Strip

PGPH-D-24-02192R2

Dear Mr. Perlman,

We are pleased to inform you that your manuscript 'Public health implications of satellite-detected widespread damage to WASH infrastructure in the Gaza Strip' has been provisionally accepted for publication in PLOS Global Public Health.

Best regards,

Nancy Angeline Gnanaselvam

Academic Editor

Kindly ensure the ethics approval document is uploaded